# Curvature Filtrations for Graph Generative Model Evaluation

**Joshua Southern**[*]
Imperial College London
jks17@ic.ac.uk

**Jeremy Wayland**[*]
Helmholtz Munich & Technical University of Munich
jeremy.wayland@tum.de

**Michael Bronstein**[†]
University of Oxford
michael.bronstein@cs.ox.ac.uk

**Bastian Rieck**[†]
Helmholtz Munich & Technical University of Munich
bastian.rieck@tum.de

## Abstract

Graph generative model evaluation necessitates understanding differences between graphs on the distributional level. This entails being able to harness salient attributes of graphs in an efficient manner. Curvature constitutes one such property that has recently proved its utility in characterising graphs. Its expressive properties, stability, and practical utility in model evaluation remain largely unexplored, however. We combine graph curvature descriptors with emerging methods from topological data analysis to obtain robust, expressive descriptors for evaluating graph generative models.

## 1  Introduction

Graph-structured data are prevalent in a number of different domains, including social networks [48], bioinformatics [38, 59], and transportation [33]. The ability to generate new graphs from a distribution is a burgeoning technology with promising applications in molecule or protein design [34, 35], as well as circuit design [47]. To compare graph generative models and advance research in this area, it is essential to have a metric that can measure the distance between a set of generated and reference graphs, with enough *expressivity* to critically evaluate model performance. Typically, this is done by using a set of descriptor functions, which map a graph to a high-dimensional representation in $\mathbb{R}^d$. An evaluator function, such as the *maximum mean discrepancy* [27, MMD], may then be used to get a distance between two distributions of graphs by comparing their vectorial representations [44, 53]. This state of the art was recently critiqued by O'Bray et al. [54] since it (i) requires numerous parameter and function choices, (ii) is limited by the *expressivity* of the descriptor function, and (iii) does not come equipped with *stability guarantees*.

We propose to overcome these issues through topological data analysis (TDA), which is capable of capturing multi-scale features of graphs while being more expressive than simple descriptor functions. TDA is built on the existence of a function of the form $f \colon V \to \mathbb{R}$, or $f \colon E \to \mathbb{R}$ on a graph $G = (V, E)$. This is used to obtain a *filtration*, i.e. an ordering of subgraphs, resulting in a set of topological descriptors, the *persistence diagrams*. Motivated by their expressive power and computational efficiency, we use recent notions of discrete curvature [19, 24, 55] to define filtrations and calculate *persistence landscapes* [10] from the persistence diagrams, thus obtaining a descriptor whose Banach space formulation permits statistical calculations. Our proposed method comes equipped with stability guarantees, can count certain substructures and measure important graph characteristics, and can be used to evaluate a variety of statistical tests since it permits computing distances between graph distributions.

---

[*]These authors contributed equally.
[†]These authors jointly directed the work.

37th Conference on Neural Information Processing Systems (NeurIPS 2023).

Our **contributions** are as follows:

- We provide a thorough theoretical analysis of the stability and expressivity of recent notions of discrete curvature, showing their fundamental utility for graph learning applications.
- Using discrete curvature and TDA, we develop a new metric for graph generative model evaluation.
- Our experiments reveal our metric is robust and expressive, thus improving upon current approaches that use simple graph descriptor and evaluator functions.

## 2  Background

The topological descriptors used in this paper are based on *computational topology* and *discrete curvature*. We give an overview of these areas and briefly comment on previous work that makes use of *graph statistics* in combination with MMD. In the following, we consider undirected graphs, denoted by $G = (V, E)$, with a set of vertices $V$ and a set of edges $E \subseteq V \times V$.

### 2.1  MMD and Metrics Based On Graph Statistics

MMD is a powerful method for comparing different distributions of structured objects. It employs *kernel functions*, i.e. positive definite similarity functions, and can thus be directly combined with standard graph kernels [8] for graph distribution comparison. However, there are subtle issues when calculating kernels in $\mathbb{R}^d$: Gaussian kernels on geodesic metric spaces, for instance, have limited applicability when spaces have non-zero curvature [23], and certain kernels in the literature are indefinite, thus violating one of the tenets of MMD [54]. Despite these shortcomings, MMD is commonly used to evaluate graph generative models [17, 71]. This is accomplished by extracting a feature vector from each graph, such as the clustering coefficient or node degree, and subsequently calculating empirical MMD values between generated and reference samples. Some works [49] combine multiple structural properties of graphs into a single metric through the Kolmorogov–Smirnov (KS) multidimensional distance [39]. Combining graph statistics into a single measure has also led to metrics between molecular graphs, such as the quantitative estimate of drug-likeness (QED), which is a common measure in drug discovery [5]. However, these simple statistics, even when considered jointly, often lack expressivity, have no stability guarantees, and their use with MMD requires numerous parameter and function choices [54].

### 2.2  Computational Topology

Computational topology assigns *invariants*—characteristic properties that remain unchanged under certain transformations—to topological spaces. For graphs, the simplest invariants are given by the 0-dimensional ($\beta_0$) and 1-dimensional ($\beta_1$) Betti numbers. These correspond to the number of connected components and number of cycles, respectively, and can be computed efficiently. Their limited expressivity can be substantially increased when paired with a scalar-valued *filtration function* $f\colon E \to \mathbb{R}$.[3] Since $f$ can only attain a finite number of values $a_0, a_1, a_2, \ldots$ on the graph, this permits calculating a *graph filtration* $\emptyset \subseteq G_0 \subseteq G_1 \ldots \subseteq G_{k-1} \subseteq G_k = G$, where each $G_i := (V_i, E_i)$, with $E_i := \{e \in E \mid f(e) \leq a_i\}$ and $V_i := \{v \in V \mid \exists e \in E_i \text{ s.t. } v \in e\}$. This *sublevel set filtration*[4] permits tracking topological features, such as cycles, via *persistent homology* [21]. If a topological feature appears for the first time in $G_i$ and disappears in $G_j$, we represent the feature as a tuple $(a_i, a_j)$, which we can collect in a *persistence diagram* $D$. Persistent homology thus tracks changes in connected components and cycles over the complete filtration, measured using a filtration function $f$. Persistence diagrams form a metric space, with the distance between them given by the *bottleneck distance*, defined as $d_B(D, D') := (\inf_{\eta\colon D \to D'} \sup_{x \in D} \|x - \eta(x)\|_\infty)$, where $\eta$ ranges over all bijections between the two diagrams. A seminal *stability theorem* [14] states that the bottleneck distance between persistence diagrams $D_f, D_g$, generated from two functions $f$ and $g$ on the same graph, is upper-bounded by $d_B(D_f, D_g) \leq \|f - g\|_\infty$. The infinity norm of the functions, a geometrical quantity, hence limits the topological variation. In practice, we convert persistence diagrams to an equivalent representation, the *persistence landscape* [10], which is more amenable to statistical analyses and the integration into machine learning pipelines.

---

[3]One can also consider functions over vertices, $f\colon V \to \mathbb{R}$, when building filtrations. See Appendix B.1 for a remark on the equivalence of these viewpoints.

[4]Swapping '$\leq$' for '$\geq$' and max for min results in the *superlevel set filtration* of equal expressivity.

**Advantageous Properties.** Persistent homology satisfies expressivity and stability properties. The choice of filtration $f$ affects expressivity: with the right filtration, persistent homology can be *more* expressive than the 1-WL test [31, 57], which is commonly used to bound the expressivity of graph neural networks [50, 52, 70]. Thus, given a suitable filtration, we can create a robust and expressive metric for comparing graphs. Moreover, as we will later see, TDA improves the expressivity of *any* $f$, meaning that even if the function on its own is not capable of distinguishing between different graphs, using it in a filtration context can overcome these deficiencies.

## 2.3   Discrete Curvature

While filtrations can be learnt [29, 31], in the absence of a well-defined learning task for graph generative model evaluation, we opt instead to employ existing functions that exhibit suitable expressivity properties. Of particular interest are functions based on *discrete curvature*, which was shown to be an expressive feature for graph and molecular learning tasks [65, 68, 69]. Curvature is a fundamental concept in differential geometry and topology, making it possible to distinguish between different types of manifolds. There are a variety of different curvature formulations with varying properties, with *Ricci curvature* being one of the most prominent. Roughly speaking, Ricci curvature is based on measuring the differences in the growth of volumes in a space as compared to a model Euclidean space. While originally requiring a smooth setting, recent work started exploring how to formulate a theory of Ricci curvature in the discrete setting [16, 19, 24, 45, 55, 62]. Intuitively, discrete curvature measures quantify a notion of similarity between node neighbourhoods, the discrete concept corresponding to 'volume.' They tend to be *larger* for structures where there are overlapping neighbourhoods such as cliques, *smaller* (or zero) for grids and *lowest* (or negative) for tree-like structures. Ricci curvature for graphs provides us with sophisticated tools to analyse the neighbourhood of an edge and recent works have shown the benefits of using some of these curvature-based methods in combination with Graph Neural Networks (GNNs) to boost performance [65], assess differences between real-world networks [61], or enable *graph rewiring* to reduce over-squashing in GNNs [64]. However, the representational power and stability properties of these measures remain largely unexplored. Subsequently, we will focus on three different types of curvature, (i) Forman–Ricci curvature [24], (ii) Ollivier–Ricci curvature [55], and (iii) Resistance curvature [19]. We find these three notions to be prototypical of discrete curvature measures, increasing in complexity and in their ability to capture *global* properties of a graph.

**Forman–Ricci Curvature.** The *Forman(–Ricci) curvature* for an edge $(i, j) \in E$ is defined as

$$\kappa_{\text{FR}}(i,j) := 4 - d_i - d_j + 3|\#_\Delta|, \tag{1}$$

where $d_i$ is the degree of node $i$ and $|\#_\Delta|$ is the number of 3-cycles (i.e. triangles) containing the adjacent nodes.

**Ollivier–Ricci Curvature.** Ollivier introduced a notion of curvature for metric spaces that measures the Wasserstein distance between Markov chains, i.e. random walks, defined on two nodes [55]. Let $G$ be a graph with some metric $d_G$, and $\mu_v$ be a probability measure on G for node $v \in V$. The *Ollivier–Ricci curvature* of *any*[5] pair $i, j \in V \times V$ with $i \neq j$ is then defined as

$$\kappa_{\text{OR}}(i,j) := 1 - \frac{1}{d_G(i,j)} W_1(\mu_i, \mu_j), \tag{2}$$

where $W_1$ refers to the first *Wasserstein distance* between $\mu_i, \mu_j$. Eq. (2) defines the Ollivier–Ricci (OR) curvature in a general setting outlined by Hoorn et al. [30]; this is in contrast to the majority of previous works in the graph setting which specify $d_G$ to be the shortest-path distance and $\mu_i, \mu_j$ to be uniform probability measures in the 1-hop neighbourhood of the node. Extending the probability measures so that they act on larger locality scales is known to be beneficial for characterising graphs [4, 26, 36]. We assume this general setting to define different notions of OR curvature on the graph, permitting us the flexibility of altering the probability measure and the metric.

**Resistance Curvature.** The resistance curvature for edges of a graph, as established in Devriendt and Lambiotte [19], is inspired by Ohm's Law and the concept of effective resistance, a well-studied, global metric between nodes in a weighted graph that quantifies the resistance exerted by the network

---

[5]In contrast to other notions of curvature, Ollivier–Ricci curvature is defined for *both* edges and non-edges.

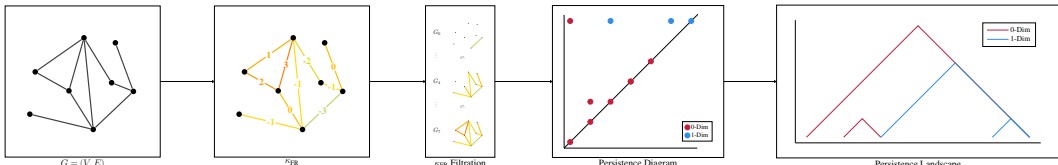

*Figure 1:* An overview of our pipeline for evaluating graph generative models using discrete curvature. We depict a graph's edges being coloured by $\kappa_{\text{FR}}$, as described in Eq. (1).The ordering on edges gives rise to a *curvature filtration*, followed by a corresponding persistence diagram and landscape. For graph generative models, we select a curvature, apply this framework element-wise, and evaluate the similarity of the generated and reference distributions by comparing their average landscapes.

when current flows between nodes. For a graph $G = (V, E)$, let $R_{ij}$ be the resistance distance between nodes $i, j \in V$, defined in Eq. (9). The *node resistance curvature* of a node $i \in V$ is then defined as $p_i := 1 - \frac{1}{2} \sum_{j \sim i} R_{ji}$. The curvature of an edge $(i, j) \in E$, what we refer to as *resistance curvature*, is then defined as

$$\kappa_{\text{R}}(i, j) := \frac{2(p_i + p_j)}{R_{ij}} \tag{3}$$

The resistance curvature of an edge is related to the average distance between the neighbourhoods of the nodes connected by the edge.

## 3 Our Method

Each notion of discrete curvature yields a scalar-valued function on the edges of a graph. We use these functions to define a family of *curvature filtrations* based on sublevel sets. In combination with persistent homology, this enables us to assess the structural properties of a given graph at multiple scales, measured via curvature. Using metrics on aggregated topological signatures—here, in the form of *persistence landscapes*—we may then compare two distributions of graphs. Specifically, we propose the following scheme for graph generative model evaluation:

1. Given a specific curvature filtration, we generate a set of *persistence diagrams* that encode the persistent homology in dimensions $0$ and $1$ for each graph in the distribution. Each diagram tracks the lifespan of connected components and cycles as they appear in the filtration of a given graph, resulting in a multi-scale summary of the graph's structure.
2. To permit an analysis on the distributional level, we convert each diagram into a more suitable representation, namely a *persistence landscape*. As functional summaries of topological information, persistence landscapes allow for easy calculation of averages and distances [10].
3. Finally, we conduct statistical comparisons between graph distributions, e.g. permutation tests using $p$-norms, in this latent space, providing an *expressive* metric for evaluating generative models.

Figure 1 illustrates our proposed pipeline using Forman–Ricci curvature. For choosing a notion of curvature in practice, implementation details, and computational performance we refer the reader to Section 3.3, Appendix A, and Appendix G respectively. We also make our framework publicly available.[6]

### 3.1 Stability

We first discuss the general stability of topological calculations, proving that changes in topological dissimilarity are bounded by changes in the filtration functions. Moreover, we show that filtrations based on discrete curvature satisfy stability properties if the underlying graph is modified.

**Topological Features.** Using the persistent homology stability theorem [14], we know that if two curvature filtrations are similar on a graph, their persistence diagrams will also be similar. However, the theorem only holds for two different functions on the *same* graph, whereas the distributional case has not yet been addressed by the literature. Our theorem provides a (coarse) upper bound.

---

[6]Source code is available at `https://github.com/aidos-lab/CFGGME`.

**Theorem 1.** *Given graphs $F = (V_F, E_F)$ and $G = (V_G, E_G)$ with filtration functions $f, g$, and corresponding persistence diagrams $D_f, D_g$, we have $d_B(D_f, D_g) \leq \max\{\mathrm{dis}(f,g), \mathrm{dis}(g,f)\}$, where $\mathrm{dis}(f,g) := |\max_{x \in E_F} f(x) - \min_{y \in E_G} g(y)|$ and vice versa for $\mathrm{dis}(g,f)$.*

This upper bound implies that changes on the level of persistence diagrams are bounded by changes between the input functions. The stability of our method thus hinges on the stability of the filtration functions, so we need to understand the behaviour of curvature under perturbations. Following previous work [54], we aim to understand and quantify the stability of our method in response to adding and deleting edges in the graph. Our stability theorems establish bounds on various discrete curvature measures for *finite*, *unweighted*, *connected* graphs in response to these perturbations. We restrict the outcome of a perturbation to graphs of the form $G' = (V, E')$ that satisfy $|E| \neq |E'|$ and do not change the number of connected components of a graph. Our theoretical results bound the new curvature $\kappa'$ according to the structural properties of $G$. For an exhaustive list of theorems and proofs, as well as experiments analysing the change in curvature for perturbed Erdős–Rényi (ER) graphs, see Appendix B and Appendix D.

**Forman–Ricci Curvature.** We first analyse Forman–Ricci curvature $\kappa_{\mathrm{FR}}$, and prove that it is stable with respect to adding and deleting edges.

**Theorem 2.** *If $G'$ is the graph generated by **edge addition**, then the updated Forman curvature $\kappa'_{FR}$ for pre-existing edges $(i,j) \in E$ can be bounded by $\kappa_{FR}(i,j) - 1 \leq \kappa'_{FR}(i,j) \leq \kappa_{FR}(i,j) + 2$. If $G'$ is the graph generated by **edge deletion**, then the updated Forman curvature $\kappa'_{FR}$ for pre-existing edges $(i,j) \in E$ can be bounded by $\kappa_{FR}(i,j) - 2 \leq \kappa'_{FR}(i,j) \leq \kappa_{FR}(i,j) + 1$.*

**Ollivier–Ricci Curvature.** Let $\mathcal{G} = (G, d_G, \mu)$ be a triple for specifying Ollivier–Ricci curvature calculations, with $G$ denoting an unweighted, connected graph, $d_G$ its associated graph metric, and $\mu := \{\mu_v \mid v \in V\}$ a collection of probability measures at each node. Furthermore, let $\delta_i$ denote the Dirac measure at node $i$ and $J(i) := W_1(\delta_i, \mu_i)$ the corresponding jump probability in the graph $G$ as defined by Ollivier [55]. Following an edge addition or deletion, we consider an updated triple $\mathcal{G}' = (G', d_{G'}, \mu')$, and remark that this yields an updated Wasserstein distance $W'_1$, calculated in terms of the new graph metric $d_{G'}$.

**Theorem 3.** *Given a perturbation (either **edge addition** or **edge deletion**) producing $\mathcal{G}'$, the Ollivier–Ricci curvature $\kappa'_{OR}(i,j)$ of a pair $(i,j)$ can be bounded via*

$$1 - \frac{1}{d_{G'}(i,j)} \left[ 2W'_{\max} + W'_1(\mu_i, \mu_j) \right] \leq \kappa'_{OR}(i,j) \leq \frac{J'(i) + J'(j)}{d_{G'}(i,j)}, \tag{4}$$

*where $J'(v) := W'_1(\delta_v, \mu'_v)$ refers to the new jump probabilities and $W'_{\max} := \max_{x \in V} W'_1(\mu_x, \mu'_x)$ denotes the maximal reaction to the perturbation (measured using the updated Wasserstein distance).*

**Resistance Curvature.** Let $G$ be an unweighted, connected graph with a resistance distance $R_{ij}$ and resistance curvature $\kappa_{\mathrm{R}}(i,j)$ for each $(i,j) \in E$. Furthermore, let $d_x$ denote the degree for node $x \in V$. We find that $\kappa_{\mathrm{R}}$ is well-behaved under these perturbations, in the sense that edge additions can only *increase* the curvature, and edge deletions can only *decrease* it. For edge additions, we obtain the following bound (see Appendix B.1.3 for the corresponding bound for edge deletions).

**Theorem 4.** *If $G'$ is the graph generated by **edge addition**, then $\kappa'_R \geq \kappa_R$, with the following bound:*

$$|\kappa'_R(i,j) - \kappa_R(i,j)| \leq \frac{\Delta_{\mathrm{add}}(d_i + d_j)}{R_{ij} - \Delta_{\mathrm{add}}}, \tag{5}$$

*where $\Delta_{\mathrm{add}} := \max_{i,j \in V} \left( R_{ij} - \frac{1}{2}\left(\frac{1}{d_i+1} + \frac{1}{d_j+1}\right)\right)$.*

> The implications of this section are that (i) the stability of our topological calculations largely hinges on the stability of the functions being used to define said filtrations, and (ii) *all* discrete curvature measures satisfy stability properties with respect to changes in graph connectivity, making curvature-based filtrations highly robust.

## 3.2 Expressivity

A metric between distributions should be non-zero when the distributions differ. For this to occur, our metric needs to be able to distinguish non-isomorphic graphs and be sufficiently expressive. Horn

et al. [31] showed that persistent homology with an appropriate choice of filtration is strictly more expressive than 1-WL, the 1-dimensional Weisfeiler–Le(h)man test for graph isomorphism. A similar expressivity result can be obtained for using resistance curvature as a node feature [65], underlining the general utility of curvature. We have the following general results concerning the expressivity or discriminative power of our topological representations.

**Theorem 5.** *Given two graphs $F = (V_F, E_F)$ and $G = (V_G, E_G)$ with scalar-valued filtration functions $f, g$, and their respective persistence diagrams $D_f, D_g$, we have $d_B(D_f, D_g) \geq \inf_{\eta:\ E_F \to E_G} \sup_{x \in E_F} |f(x) - g(\eta(x))|$, where $\eta$ ranges over all maps from $E_F$ to $E_G$.*

Theorem 5 implies that topological distances are generally more discriminative than the distances between the filtration functions. Thus, calculating topological representations of graphs based on a class of functions improves discriminative power. To further understand the expressive power of curvature filtrations, we analyse strongly-regular graphs, which are often used for studying GNN expressivity as they cannot be distinguished by $k$-WL, the $k$-dimensional Weisfeiler–Le(h)man test, if $k \leq 3$ [2, 7, 52]. Additionally, we explore how curvature filtrations can count substructures, an important tool for evaluating and comparing expressivity [56]. To the best of our knowledge, ours is the first work to explore discrete curvature and curvature-based filtrations in this context.

**Distinguishing Strongly-Regular Graphs.** Strongly-regular graphs are often used to assess the expressive power of graph learning algorithms, constituting a class of graphs that are particularly hard to distinguish. We briefly recall some definitions. A connected graph $G$ with diameter $D$ is called *distance regular* if there are integers $b_i, c_i, (0 \leq i \leq D)$ such that for any two vertices $x, y \in V$ with $d(x, y) = i$, there are $c_i$ neighbours of $y$ in $k_{i-1}(x)$ and $b_i$ neighbours of $y$ in $k_{i+1}(x)$. For a distance-regular graph, the intersection array is given by $\{b_0, b_1, \ldots, b_{D-1}; c_1, c_2, \ldots, c_D\}$. A graph is called *strongly regular* if it is distance regular and has a diameter of 2 [18]. We first state two theoretical results about curvature.

**Theorem 6** (Expressivity of curvature notions)**.** *Both Forman–Ricci curvature and Resistance curvature* cannot *distinguish distance-regular graphs with the same intersection array, whereas Ollivier–Ricci curvature* can *distinguish the Rook and Shrikhande graphs, which are strongly-regular graphs with the same intersection array.*

The Rook and Shrikhande graph *cannot* be distinguished by 2-WL [9, 52]. However, OR curvature is sensitive to differences in their first-hop peripheral subgraphs [22], thus distinguishing them. This result shows the limitations of Forman–Ricci and Resistance curvature, as well as the benefits of using Ollivier–Ricci curvature. We show further experiments with curvature notions on strongly-regular graphs in the experimental section, observing improvements whenever we use them as filtrations.

**Counting Substructures.** Evaluating the ability of curvature to encode structural information is a crucial aspect for understanding its expressivity and validating its overall utility in graph learning. It has previously been shown that incorporating structural information of graphs can extend the expressive power of message-passing neural networks [11, 25, 42]. Additionally, Bouritsas et al. [9] show how GNNs can be (i) strictly more expressive than 1-WL by counting local isomorphic substructures (e.g. cliques), and (ii) exhibit predictive performance improvements when adding such substructure counts to the node features. For instance, some strongly-regular graphs can be distinguished by counting 4-cliques. Discrete curvature measures are informed by these local substructures and have been shown to improve expressivity beyond 1-WL [65] when included as a node feature. Nevertheless, there is limited work exploring what substructure information curvature carries, which would allow us to describe the expressivity of the measure. Moreover, persistent homology can track the number of cycles over the filtration of interest, allowing additional structural information to be encoded at multiple scales. Thus, we will explore the extent to which substructures can be counted for different curvatures (with and without a topological component) in a subsequent experiment, providing evidence on the expressive power of curvature filtrations. See Table 4 for our experimental results, and Appendix E for a more detailed discussion on the tendencies of each curvature notion when counting substructures.

### 3.3 Choosing a Curvature Notion in Practice

In Section 3.1, we showed that all three of our prototypical curvature notions exhibit advantageous stability properties, thus implying that they may all be used *reliably* within our method to measure

the distance between two sets of graphs. However, which curvature should be chosen in practice? In general, we find that the answer to this question lies at the intersection of *expressivity* and *scalability*, but depends ultimately on the nature of the experiment at hand. Nevertheless, we aim to provide an intuition for all three notions, along with a general recommendation. We hope this, in conjunction with the experimental and computational complexity results in Section 4 and Appendix G will help practitioners in making an optimal choice.

**Comparison.** Forman–Ricci is arguably the simplest and most local notion of curvature. Though limited in expressivity when compared to Ollivier–Ricci curvature, it is by far the most efficient to compute. Resistance curvature, by contrast, is the most global notion, making it sensitive to large substructures. However, computing the effective resistance metric on a graph requires inverting the Laplacian, making it far less efficient than Forman and Ollivier–Ricci, especially for large graphs. Finally, we have Ollivier–Ricci curvature, which we have found to be the most expressive based on its ability to (i) distinguish strongly-regular graphs, and (ii) count substructures. It is also the most versatile, given the option to adapt the underlying probability measure; this comes at the cost of lower computational performance in comparison to Forman—Ricci curvature, though.

> Given its high expressivity, as well as its overall experimental and computational performance, we recommend using **Ollivier–Ricci** curvature whenever feasible.

## 4 Experiments

We have proven the theoretical stability of discrete curvature notions under certain graph perturbations. We also illustrated their ability to distinguish distance-regular and strongly-regular graphs. Subsequently, we will discuss empirical experiments to evaluate these claims and to further test the utility of our methods.

### 4.1 Distinguishing Strongly-Regular Graphs

In addition to the theoretical arguments outlined, we explore the ability of our method to distinguish strongly-regular graphs in a subset of data sets, i.e. sr16622, sr261034, sr281264, and sr401224. These data sets are known to be challenging to classify since they cannot be described in terms of the 1-WL test [7]. Our main goal is *not* to obtain the best accuracy, but to show how the discriminative power of discrete curvature can be improved by using it in a filtra-

*Table 1:* Success rate ($\uparrow$) of distinguishing pairs of strongly-regular graphs when using either raw curvature values or a curvature filtration.

| Method | sr16622 | sr261034 | sr281264 | sr401224 |
|---|---|---|---|---|
| $\kappa_{\text{FR}}$ | 0.00 | 0.00 | 0.00 | 0.00 |
| $\kappa_{\text{OR}}$ | **1.00** | 0.78 | **1.00** | 0.00 |
| $\kappa_{\text{R}}$ | 0.00 | 0.00 | 0.00 | 0.00 |
| Filtration ($\kappa_{\text{FR}}$) | **1.00** | 0.20 | 0.00 | **0.93** |
| Filtration ($\kappa_{\text{OR}}$) | **1.00** | **0.89** | **1.00** | **0.93** |
| Filtration ($\kappa_{\text{R}}$) | **1.00** | 0.20 | 0.00 | **0.93** |

tion context. Table 1 depicts the results of our classification experiment. We perform a pairwise analysis of *all graphs* in the data set, calculating distances based on histograms of discrete curvature measurements, or based on the bottleneck distance between persistence diagrams ('Filtrations'). Subsequently, we count all non-zero distances ($> 1 \times 10^{-8}$ to correct for precision errors). Our main observation is that combining TDA with curvature is always better than or equal to curvature without TDA. Similar to our theoretical predictions, both resistance curvature and Forman curvature fail to distinguish any of the graphs without TDA. We therefore show the benefits from an expressivity point of view for using discrete curvature as a filtration. Notably, we achieve the best results with OR curvature, which is particularly flexible since it permits changing the underlying *probability measure*. Using a probability measure based on random walks (see Appendix F) takes into account higher-order neighbourhoods and improves discriminative power (on sr261034, the pairwise success rate drops to 0.0/0.2 with raw/TDA values, respectively, if the uniform probability measure is used).

### 4.2 Expressivity experiments with the BREC dataset

We evaluate discrete curvatures and their filtrations on the BREC data set, which was recently introduced to evaluate GNN expressiveness [66]. The data set consists of different categories of graph pairs (Basic, Regular, and Extension), which are distinguishable by 3-WL but not by 1-WL, as well as Strongly-Regular (STR) and CFI graph pairs, which are indistinguishable using 3-WL. We explore the ability of curvature filtrations to distinguish these graph pairs and compare them

*Table 2:* Success rate (↑) of distinguishing pairs of graphs in the BREC dataset when using different discrete curvatures and their filtrations.

| Method | Basic (56) | Regular (50) | STR (50) | Extension (97) | CFI (97) |
|---|---|---|---|---|---|
| 1-WL | 0.00 | 0.00 | 0.00 | 0.00 | 0.00 |
| 3-WL | **1.00** | **1.00** | 0.00 | **1.00** | **0.59** |
| $S_3$ | 0.86 | 0.96 | 0.00 | 0.05 | 0.00 |
| $S_4$ | 1.00 | 0.98 | **1.00** | 0.84 | 0.00 |
| $\kappa_{OR}$ | 1.00 | 0.96 | 0.06 | 0.93 | 0.00 |
| $\kappa_{FR}$ | 0.96 | 0.92 | 0.00 | 0.52 | 0.00 |
| $\kappa_R$ | **1.00** | **1.00** | 0.00 | **1.00** | 0.04 |
| Filtration ($\kappa_{OR}$) | **1.00** | **1.00** | 0.08 | 0.95 | 0.00 |
| Filtration ($\kappa_{FR}$) | 0.98 | 0.96 | 0.04 | 0.61 | 0.00 |
| Filtration ($\kappa_R$) | **1.00** | **1.00** | 0.04 | **1.00** | 0.04 |

to substructure counting, $S_3$ and $S_4$, which involves enumerating all 3-node/4-node substructures around nodes in combination with the WL algorithm. These approaches, unlike discrete curvature, have limited practical applications due to their high computational complexity. Similar to our experiments on strongly-regular graphs, we calculate Wasserstein distances based on histograms of OR curvature measurements between the pairs of graphs. Subsequently, we count all non-zero distances ($> 1 \times 10^{-8}$ to correct for precision errors). Our main observations from Table 2 are that curvature *can* distinguish graphs which are 3-WL indistinguishable. Additionally, we observe improvements in success rate using the filtration on the Basic, Regular, STR and Extension graph pairs. Moreover, our curvature-based approach performs competitively and sometimes even better than $S_4$, which has been shown to be extremely effective in graph learning tasks [9]. Despite its empirical prowess, $S_4$ is computationally expensive, making it an *infeasible* measure in many applications. Discrete curvature and its filtrations, by contrast, scale significantly better.

## 4.3 Behaviour with Respect to Perturbations

To explore the behaviour of curvature descriptors under perturbations, we analyse the correlation of our metric when adding and removing edges in the 'Community Graph' data set: we increase the fraction of edges added or removed from $0.0$ to $0.95$, measuring the distance between the perturbed graphs and the original graphs for each perturbation level. Following O'Bray et al. [54], we require a suitable distance measure to be highly correlated with increasing amounts of perturbation. We compare to current approaches that use descriptor functions with MMD. As Table 3 shows, a curvature filtration yields a higher correlation than curvature in

*Table 3:* Pearson correlation (↑) of measures when adding/removing edges.

| Measure | Adding Edges | Removing Edges |
|---|---|---|
| Laplacian | $0.457 \pm 0.013$ | $0.420 \pm 0.000$ |
| Clust. Coeff. | $0.480 \pm 0.012$ | $0.504 \pm 0.020$ |
| Degrees | $0.761 \pm 0.003$ | $0.995 \pm 0.000$ |
| $\kappa_{FR}$ | $0.420 \pm 0.000$ | $0.432 \pm 0.003$ |
| $\kappa_{OR}$ | $0.903 \pm 0.005$ | $0.910 \pm 0.002$ |
| $\kappa_R$ | $0.420 \pm 0.004$ | $0.441 \pm 0.005$ |
| Filtration ($\kappa_{FR}$) | $0.571 \pm 0.006$ | $0.996 \pm 0.006$ |
| Filtration ($\kappa_{OR}$) | $\mathbf{0.997 \pm 0.000}$ | $\mathbf{0.970 \pm 0.005}$ |
| Filtration ($\kappa_R$) | $0.730 \pm 0.005$ | $0.954 \pm 0.008$ |

combination with MMD, showing the benefits of employing TDA from a stability perspective. Additionally, curvature filtrations improve upon the normalized Laplacian and clustering coefficient (again, we used MMD for the comparison of these distributions). OR curvature exhibits particularly strong results when adding/removing edges, even surpassing the local degrees of a graph (which, while being well-aligned with perturbations of the graph structure, suffer in terms of overall expressivity).

## 4.4 Counting Substructures

The ability of a descriptor to count local substructures is important for evaluating its expressive power [3]. We follow Chen et al. [12], who assess the ability of GNNs to count substructures such as triangles, chordal cycles, stars and tailed triangles. This is achieved by generating regular graphs with random edges removed, and counting the number of occurrences of each substructure in a given graph. To assess the power of curvature to capture such local structural features, we use the same experimental setup and pass the edge-based curvatures through a simple 1-layer MLP to output the substructure count. Additionally, we evaluate the effect of using curvature as a filtration.

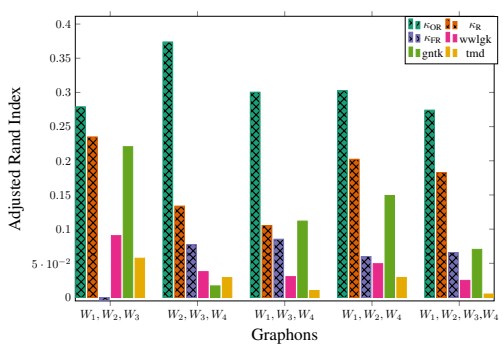
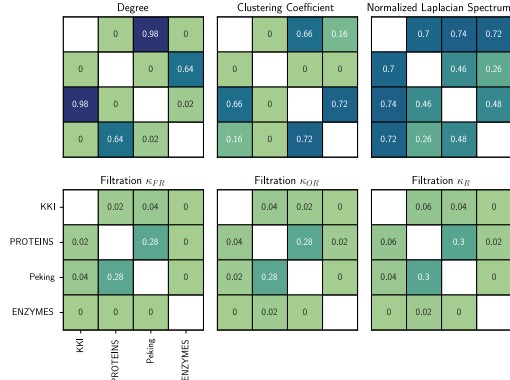

*(a)* Curvature distinguishes graphon data sets    *(b)* Curvature distinguishes bioinformatics data sets

*Figure 2:* (a) Adjusted Rand Index (↑) for clustering sets of four graphons. We compare our curvature filtrations (⊠) to kernel-based methods. (b) Permutation testing values (↓) for distinguishing different bioinformatics data sets. Position $(i, j)$ in each matrix denotes a permutation test between data set $i$ and data set $j$.

*Table 4:* MAE (↓) for counting substructures based on raw curvature values and curvature filtrations. The 'Trivial Predictor' always outputs the mean training target.

| Method | Triangle | Tailed Tri. | Star | 4-Cycle |
|---|---|---|---|---|
| Trivial Predictor | 0.88 | 0.90 | 0.81 | 0.93 |
| GCN | 0.42 | 0.32 | **0.18** | **0.28** |
| $\kappa_{FR}$ | 0.54 | 0.56 | 0.72 | 0.53 |
| $\kappa_{OR}$ | 0.33 | 0.31 | 0.40 | 0.31 |
| $\kappa_{R}$ | 0.59 | 0.50 | 0.72 | 0.47 |
| Filtration ($\kappa_{FR}$) | 0.45 | 0.52 | 0.49 | 0.60 |
| Filtration ($\kappa_{OR}$) | **0.23** | **0.24** | 0.34 | 0.31 |
| Filtration ($\kappa_{R}$) | 0.47 | 0.48 | 0.36 | 0.42 |

Table 4 shows our experimental results, reported on the pre-defined test split defined by Chen et al. [12]. In comparison to Graph Convolutional Networks [40], we find that OR curvature exhibits improved performance in counting small local structures such as triangles and tailed triangles; this is a surprising finding given the smaller computational footprint of this curvature formulation (in comparison to a GCN). Moreover, we find that the OR curvature performs better than both the Forman curvature and the resistance curvature for the different substructures in the data. We also observe that combining curvature with TDA almost always improves upon using the curvature alone, the only exception being Forman curvature for counting 4-cycles. We leave a more detailed investigation of these phenomena for future work.

## 4.5 Synthetic Graph Generative Model Evaluation

To have a ground truth for a graph distribution, we tested our metric's ability to *distinguish graphons*. Following the approach suggested by Sabanayagam et al. [60], we generate four graphons, $W_1(u, v) = uv$, $W_2(u, v) = \exp\{-\max(u, v)^{0.75}\}$, $W_3(u, v) = \exp\{-0.5 * (\min(u, v) + u^{0.5} + v^{0.5})\}$ and $W_4(u, v) = \|u - v\|$. Sampling from these graphons produces dense graphs, and we control their size to be between 9 and 37 nodes, thus ensuring that we match the sizes of molecular graphs in the ZINC data set [32], an important application for generative models.

We perform experiments by considering all combinations of three and four graphons. We generate distances between graphs with our method as well as other kernel-based approaches, and use spectral clustering to separate the distributions. We measure the performance of the algorithms using the Adjusted Rand Index (ARI) of the predicted clusters, comparing to three state of the art, kernel-based approaches: (i) Wasserstein Weisfeiler–Le(h)man graph kernels [63], (ii) graph neural tangent kernels [20], and (iii) Tree Mover's Distance [13]. From Figure 2a, we find that a filtration based on OR curvature is better able to distinguish and cluster graphons than the previously-described approaches based on kernels and it performs best for all sets of graphons. We also observe that OR curvature performs better than other discrete curvatures, with resistance curvature achieving higher ARI than Forman curvature. Notice that unlike these kernel approaches, our method can be easily extended to provide a proper metric between distributions of graphs.

### 4.6 Real-World Graph Generative Model Evaluation

By converting the generated persistence diagrams into a *persistence landscape*, we can generate an average topological descriptor for all graphs in a distribution [10]. This allows us to calculate norms between graph distributions, making it possible to perform two-sample and permutation testing, unlike a majority of kernel-based approaches, which provide a distance between *all* individual graphs, or would require MMD to assess mean similarities. We randomly sample ten graphs from four different bioinformatics data sets, i.e. KKI, PROTEINS, Peking, and ENZYMES [51]. We measure the distance between two data sets using the $L^P$ norm between their average persistence landscapes. We then permute graphs from both samples, randomly selecting sets of equal size, measure their respective distances, and finally aggregate the fraction of distances which are higher than the original. A low fraction indicates that distances are *lower* for permutations than between the original sets of graphs, suggesting that the metric can distinguish between the two distributions. We compare our approach to previous methods that combine graph statistics with MMD. Using a significance threshold of $p < 0.05$, we see in Figure 2b that both Forman and the OR curvature are able to distinguish *all but one pair of data sets, an improvement over all the other approaches*. In general, we find that fractions are lower using curvature filtrations than graph statistic based approaches, demonstrating the utility of our approach.

## 5 Conclusion

We have described the first thorough analysis of both *stability* and *expressivity* of discrete curvature notions and their filtration formulations on graphs. We believe this to be important for the community in multiple contexts, ranging from improving expressivity of GNNs to understanding the robustness of curvature on graphs. Using curvature filtrations and their topological descriptors (here: *persistence landscapes*), we develop a new metric to measure distances between graph distributions. Our metric can be used for evaluating graph generative models, is robust and expressive, and improves upon current approaches based on graph descriptor and evaluator functions. We have also demonstrated clear advantages over state-of-the-art methods that combine graph statistics with MMD, providing instead a metric with (i) well-understood parameter and function choices, (ii) stability guarantees, (iii) added expressivity, and (iv) improved computational performance. Most notably, we scale *significantly* better for large populations of molecular-sized graphs (see Appendix G for more details), which we consider crucial for current graph generative model applications. We hope that our pipeline will provide a principled, interpretable, and scalable method for practitioners to use when evaluating the performance of graph generative models.

Future work could explore other representations [1, 58], focus on different filtration constructions [15], new curvature measures, or further extend our stability and expressivity results (for instance to the setting of weighted graphs). Given the beneficial performance and flexibility of Ollivier–Ricci curvature in our experiments, we believe that changing—or *learning*—the probability measure used in its calculation could lead to further improvements in terms of expressivity, for example. Another relevant direction involves incorporating node and edge features into the distance measure and applying the model to specific use cases such as evaluating molecule generation. We envision that this could be done using bi-filtrations or by considering node features for the curvature calculations.

## Acknowledgments and Disclosure of Funding

The authors want to thank the anonymous reviewers for their comments, which helped us improve the paper. We are also grateful for the area chair who believed in this work. The authors are grateful to Fabrizio Frasca and Leslie O'Bray for valuable feedback on early versions of the manuscript. J.S. is supported by the UKRI CDT in AI for Healthcare http://ai4health.io (Grant No. P/S023283/1). M.B. is supported in part by the ERC Consolidator grant No. 724228 (LEMAN) and the EPSRC Turing AI World-Leading Researcher Fellowship. B.R. is supported by the Bavarian State government with funds from the *Hightech Agenda Bavaria*. He wishes to dedicate this paper to his son Andrin.

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

# A  Pseudocode

Here we give pseudocode for various parts of the method, highlighting the most relevant aspects of using curvature filtrations to evaluate graph generative models. First, we outline a crucial part of our evaluation framework in Algorithm 1: how we compute summary topological descriptors for sets of graphs. This algorithm, taken from Bubenik [10], assumes a list of precomputed persistence landscapes, one for each graph. Algorithm 2, on the other hand, outlines our procedure for generating a distance between two sets of graphs using their summary topological descriptors.

---

**Algorithm 1** Compute Average of Persistence Landscapes

---

**Require:** $\Lambda$ is a list of persistence landscapes
**Ensure:** $\bar{\Lambda}$ is the average persistence landscape
  **function** COMPUTEAVERAGELANDSCAPE($\Lambda$)
    $n \leftarrow$ length of $\Lambda$
    $D \leftarrow$ maximum homology degree occurring in $\Lambda$
    $\bar{\Lambda} \leftarrow$ empty persistence landscape with $D$ homology dimensions.
    $\lambda^i(k,t) \leftarrow$ the piecewise-linear function encoding hom-deg $k$ contained in $\lambda^i \in \Lambda$.
    $R \leftarrow$ maximal domain for each function contained in $\Lambda$.
    $k \leftarrow 0$
    **while** $k \leq D$ **do**
      **for** $t \in R$ **do**
        $\bar{\Lambda}(k,t) \leftarrow \frac{1}{n} \sum_i^n \lambda^i(k,t)$
      **end for**
      $k \leftarrow k+1$
    **end while**
    **return** $\bar{\Lambda}$
  **end function**

---

Recall that persistence landscapes are *a collection of piecewise linear functions* that encode the homological information tracked by a specified filtration. Thus, to compute the average landscape $\bar{\Lambda}$ at each dimension, we simply sum the piecewise linear functions, and divide by the total number of landscapes in consideration. To understand our method for comparing sets of graphs, and thus evaluating graph generative models, based on average persistence landscapes see Algorithm 2.

# B  Additional Proofs on Stability and Expressivity

## B.1  Stability Proofs

**Theorem 1.** *Given graphs $F = (V_F, E_F)$ and $G = (V_G, E_G)$ with filtration functions $f, g$, and corresponding persistence diagrams $D_f, D_g$, we have $d_B(D_f, D_g) \leq \max\{\mathrm{dis}(f,g), \mathrm{dis}(g,f)\}$, where $\mathrm{dis}(f,g) := |\max_{x \in E_F} f(x) - \min_{y \in E_G} g(y)|$ and vice versa for $\mathrm{dis}(g,f)$.*

*Proof.* Considering the calculation of persistence diagrams based on scalar-valued filtrations functions, every point in the persistence diagram $D_f$ can be written as a tuple of the form $(f(e_F), f(e'_F))$, with $e_F, e'_F \in E_F$; the sample applies for $D_g$. The inner distance between such tuples that occur in the bottleneck distance calculation can thus be written as

$$\|(f(e_F), f(e'_F)) - (g(e_G), g(e'_G))\|_\infty. \tag{6}$$

The maximum distance that can be achieved using this expression is determined by the maximum variation of the functions, expressed via $\mathrm{dis}(f,g)$ and $\mathrm{dis}(g,f)$, respectively. $\square$

**Edge Filtrations versus Vertex Filtrations**  Our results are structured to address filtrations built by a function on the *edges* of a graph $G = (V, E)$, $f \colon E \to \mathbb{R}$. This matches our notions of discrete curvature, which are also defined edge-wise. $f$ gives an explicit ordering on $E$ and thus an induced ordering on $V$ given by:

$$v \leq v' \iff \sum_{e \in E_v} f(e) \leq \sum_{e' \in E_{v'}} f(e')$$

---

**Algorithm 2** Computing Distance between Two Sets of Graphs

---

**Require:** $\mathcal{G}_1$ is a set of graphs.
**Require:** $\mathcal{G}_2$ is a set of graphs.
**Require:** $D$ to be the maximum homology degree to be computed.
**Require:** $f$ is a function to generate a filtration for persistent homology computations.
**Ensure:** $Dist$ is the distance between $\mathcal{G}_1, \mathcal{G}_2$.

   **function** COMPUTESETDISTANCE($\mathcal{G}_1, \mathcal{G}_2$)
      $\Lambda_1 \leftarrow$ is an empty list.
      $\Lambda_2 \leftarrow$ is an empty list.
      $DistVec \leftarrow$ is an empty $D+1$-dimensional vector.
      **for** $G \in \mathcal{G}_1$ **do**
         $L \leftarrow$ ComputePersistenceLandscape($G, f$)
         $AddItem(\Lambda_1, L)$
      **end for**
      **for** $G \in \mathcal{G}_2$ **do**
         $L \leftarrow$ ComputePersistenceLandscape($G, f$)
         $AddItem(\Lambda_2, L)$
      **end for**
      $\bar{\Lambda}_1 \leftarrow$ ComputeAverageLandscape($\Lambda_1$)
      $\bar{\Lambda}_2 \leftarrow$ ComputeAverageLandscape($\Lambda_1$)         ▷ By Algorithm 1.
      $k \leftarrow 0$
      **while** $k \leq D$ **do**
         $DistVec[k] \leftarrow |supNorm(\bar{\Lambda}_1(k,t)) - supNorm(\bar{\Lambda}_2(k,t))|$
      **end while**
      $Dist \leftarrow ||DistVec||_2$
      **return** $Dist$
   **end function**

   **function** COMPUTEPERSISTENCELANDSCAPE($G, f, D$)
      $P \leftarrow$ compute persistence diagram for $G$ using $f$ for homology degree $k \in \{0, \cdots, D\}$.
      $L \leftarrow$ transform $P$ into a persistence landscape.     ▷ See [10] for implementation.
      **return** $L$
   **end function**

---

where $E_x$ is the set of edges incident to $x \in V$. However, one can also define a filtration directly over *vertices* with a scalar valued function $h \colon V \to \mathbb{R}$. By assumption, $h$ can attain only a finite number of values , call the unique values $b_1, b_2, \ldots b_k$. Thus, we can also compute a filtration $\emptyset \subseteq G_0 \subseteq G_1 \ldots \subseteq G_{k-1} \subseteq G_k = G$, where each $G_i := (V_i, E_i)$, with $V_i := \{v \in V \mid h(v) \leq b_i\}$ and $E_i := \{e \in E \mid \max_{v \in e} h(v) \leq b_i\}$. Similarly, the explicit ordering on $V$ given by $h$ induces an ordering on $E$:

$$e \leq e' \iff \max_{v \in e} h(v) \leq \max_{v' \in e'} h(v')$$

The key idea here is that *either choice gives rise to an ordering of both edges, and vertices* that are used to calculate persistent homology of the graph. This means that the arguments for Theorem 1 and Theorem 5 also bound the bottle-neck distance for persistence diagrams generated using vertex filtrations.

**Graph perturbations.** Here we explicitly specify a common framework used in the proofs for stability of curvature functions. As mentioned in the main text, we consider perturbations to *unweighted*, *connected* graphs $G = (V, E)$, with $|V| = n$ and $|E| = m$. In the case of *edge addition*, let $i*$ and $j*$ be arbitrary vertices that we wish to connect with a *new* edge, forming our new graph $G' = (V, E')$ where $E' = E \cup (i*, j*)$ such that $|E'| = m + 1$. For *edge deletion*, we similarly let $(i*, j*) \in E$ be the edge we delete such that $E' \subset E$ and $|E'| = m - 1$. Moreover, we only consider edges $(i*, j*)$ that leave $G'$ connected.

### B.1.1 Forman–Ricci Curvature

**Theorem 2.** *If $G'$ is the graph generated by **edge addition**, then the updated Forman curvature $\kappa'_{FR}$ for pre-existing edges $(i, j) \in E$ can be bounded by $\kappa_{FR}(i, j) - 1 \leq \kappa'_{FR}(i, j) \leq \kappa_{FR}(i, j) + 2$. If $G'$ is the graph generated by **edge deletion**, then the updated Forman curvature $\kappa'_{FR}$ for pre-existing edges $(i, j) \in E$ can be bounded by $\kappa_{FR}(i, j) - 2 \leq \kappa'_{FR}(i, j) \leq \kappa_{FR}(i, j) + 1$.*

*Proof.* We first handle the case of **edge addition**, using the graph perturbation framework specified above. By definition $\kappa_{FR}(i, j)$ depends *only* on the degrees of the source and target $(i, j) \in E$ and the number of triangles formed using $(i, j)$, $|\#_{\Delta_{ij}}| = |N(i) \cap N(j)|$, where $N(i), N(j)$ are the set of neighbouring nodes for $i, j$ respectively. This is a local computation– all relevant information can be computed in the subgraph surrounding the inserted edge $(i*, j*)$. Thus, in order to understand stability of $\kappa_{FR}(i, j)$, we need to understand how $N(i)$ and $N(j)$ change under graph perturbations. For our new graph $G'$, the only edges with potential to change their curvature lie in the set:

$$E_* := \{(u, v) \in E | u, v \in N(i*) \cup N(j*)\}$$

For the new edge $(i*, j*) \in E_*$, we can directly compute $\kappa_{FR}(i*, j*)$ based on the original structure of the graph. However, in terms of stability we are interested in the other members of $E_*$, i.e. edges in the original graph. The analysis of $E_*$ can be split into two cases: one of the nodes is $i*$ or $j*$ or neither is.

*Case 1*: WLOG assume $(i, j) = (i*, j) \in E_*$. Clearly, $d'_i = d_i + 1$. As for $|\#'_{\Delta_{ij}}|$, this can maximally be increased by 1 in the case that $j* \in N(j)$, else the triangle count stays the same.

*Case 2*: Let $(i, j) \in E_*$ where $i, j \in V \setminus \{i*, j*\}$. In this case, there is no change to the degree nor the number $|\#'_{\Delta_{uv}}|$.

Thus *Case 1* defines the bounds which are dictated as follows: if $(i*, j*)$ forms a new triangle, our curvature can increase by 2, and if no triangle is formed the curvature can decrease by 1 in response to the increased degree. Thus we can bound $\kappa'_{FR}(i, j) := 4 - d'_i - d'_j + 3|\#'_{\Delta_{ij}}|$ as follows:

$$\kappa_{FR}(i, j) - 1 \leq \kappa'_{FR}(i, j) \leq \kappa_{FR}(i, j) + 2$$

The case of **edge deletion** can be handled similarly. Again, we need only consider the edges in $E_*$, as defined in the proof above, and can make the same case argument.

*Case 1* WLOG assume $(i, j) = (i*, j) \in E_*$. Clearly, $d'_i = d_i - 1$. As for $|\#'_{\Delta_{ij}}|$, this can maximally be decreased by 1.

*Case 2*: Let $(i, j) \in E_*$ where $i, j \in V \setminus \{i*, j*\}$. Degree and number of triangles do not change in response to the perturbation.

Again, *Case 1* gives rise to the following bounds hold for $\kappa'_{FR}$:

$$\kappa_{FR}(i, j) - 2 \leq \kappa'_{FR}(i, j) \leq \kappa_{FR}(i, j) + 1$$

$\square$

### B.1.2 Ollivier–Ricci Curvature

The definition of $\kappa_{OR}$ establishes a relationship between the graph metric $d_G$, the Wasserstein distance $W_1$, the probability distributions $\mu_i, \mu_j$ at nodes $i, j$ and the curvature. Given that we are considering unweighted, and connected graphs we know that $(V, d_G)$ is a well-defined metric space and therefore $W_1$ (as defined in [55]) defines the $L_1$ transportation distance between two probability measures $\mu_i, \mu_j$ with respect to the metric $d_G$. This is relevant for a much larger class of graph metrics than just the standard choice of the shortest path distance. We use results from [55] and the metric properties of $W_1$ and $d_G$ on graphs to bound the potential changes in $\kappa_{OR}$ following an edge perturbation.

**Lemma 1.** *Consider the triple $\mathcal{G} = (G, d_G, \mu)$. Let $\delta_i$ denote the Dirac measure at node $i$ and $J(i) := W_1(\delta_i, \mu_i)$ the corresponding jump probability in the graph G. The Ollivier–Ricci curvature*

$\kappa_{OR}(i, j)$ *satisfies the following Bonnet-Myers inspired upper bound:*

$$\kappa_{OR}(i, j) \leq \frac{J(i) + J(j)}{d_G(i, j)} \tag{7}$$

*Proof.* Rearranging the original definition for OR curvature gives:

$$W_1(\mu_i, \mu_j) = d_G(i, j)(1 - \kappa_{OR}(i, j))$$

By definition of the $W_1$, we have $d_G(i, j) = W_1(\delta_i, \delta_j)$. Using this and the fact that $W_1$ satisfies the triangle inequality property, we can construct the desired upper bound on $\kappa_{OR}$:

$$d_G(i, j) \leq W_1(\delta_i, \mu_i) + W_1(\mu_i, \mu_j) + W_1(\delta_j, \mu_j)$$
$$d_G(i, j) \leq J(i) + d_G(i, j)(1 - \kappa_{OR}(i, j)) + J(j)$$
$$d_G(i, j)(1 - (1 - \kappa_{OR}(i, j))) \leq J(i) + J(j)$$
$$\kappa_{OR}(i, j) \leq \frac{J(i) + J(j)}{d_G(i, j)}$$

$\square$

**Theorem 3.** *Given a perturbation (either **edge addition** or **edge deletion**) producing $\mathcal{G}'$, the Ollivier–Ricci curvature $\kappa'_{OR}(i, j)$ of a pair $(i, j)$ can be bounded via*

$$1 - \frac{1}{d_{G'}(i, j)}\left[2W'_{\max} + W'_1(\mu_i, \mu_j)\right] \leq \kappa'_{OR}(i, j) \leq \frac{J'(i) + J'(j)}{d_{G'}(i, j)}, \tag{4}$$

*where $J'(v) := W'_1(\delta_v, \mu'_v)$ refers to the new jump probabilities and $W'_{\max} := \max_{x \in V} W'_1(\mu_x, \mu'_x)$ denotes the maximal reaction to the perturbation (measured using the updated Wasserstein distance).*

*Proof.* We first prove the *upper bound*. Given that $G'$ is still connected (by assumption), and both $W'_1$ and $d_{G'}$ still satisfy the metric axioms, this result follows directly from Lemma 1. For proving the *lower bound*, recall from Section 3.1 that $\mathcal{G}' = (G', d_{G'}, \mu')$ specifies the behaviour of the new graph metric $d_{G'}$ and the and the updated probability measure $\mu'$ in response to the perturbation. Moreover, this defines a new Wasserstein distance $W'_1$ and we will show that the maximum reaction (as evaluated by $W'_1$) to the perturbation $W'_{\max} := \max_{x \in V} W'_1(\mu'_x, \mu_x)$ can be used to express a general lower bound for OR curvature in the event of a perturbation. As per Eq. (2), we can define our curvature following the perturbation as:

$$\kappa'_{OR}(i, j) = 1 - \frac{1}{d_{G'}(i, j)}W'_1(\mu'_i, \mu'_j) \tag{8}$$

Once again, we can make use of the metric properties of $W'_1$, to establish the lower bound as

$$\kappa'_{OR}(i, j) \geq 1 - \frac{1}{d_{G'}(i, j)}\left[W'_1(\mu_i, \mu'_i) + W'_1(\mu_j, \mu'_j) + W'_1(\mu_i, \mu_j)\right]$$
$$\geq \frac{1}{d_{G'}(i, j)}\left[2W'_{\max} + W'_1(\mu_i, \mu_j)\right].$$

$\square$

### B.1.3 Resistance Curvature

**A brief clarification on inverting edge weights.** The common practice when computing effective resistance is to invert the edge weights of a graph in order to get a resistance. Given the spirit of resistance from circuit theory, we know that a high resistance should make it difficult for current to pass between nodes. Analogously when thinking about our graph as a markov chain, this would correspond to a low transition probability. So, if we think about our edge weights as coming from some kernel where higher similarity results in a higher edge weight, then we should definitely invert our edge weights to get to resistance. However, in the case that our edge weights represent the cost of travelling between nodes, then this is a suitable proxy for resistance in which case inverting the nodes is unnecessary. In order to achieve the theoretical properties of curvature with well known examples described in [19], we *do not* invert the edge weights in our experiments. Which means

that the curvature itself interprets the edge weights themselves as a cost/resistance; we think it is an important point to specify especially given the similarity to markov chains and the borrowed terminology from circuit theory.

**Definitions.** The resistance distance, intuitively, measures how well connected two nodes are in a graph. It is defined in [19] as:

$$R_{ij} := (\mathbf{e}_i - \mathbf{e}_j)^\mathsf{T} Q^\dagger (\mathbf{e}_i - \mathbf{e}_j) \tag{9}$$

Here $Q$ is the normalized laplacian (weighted degrees on the diagonal, see [19]), $Q^\dagger$ the Moore-Penrose inverse, and $\mathbf{e}_i$ is $i^{th}$ unit vector. This is the main feature that will be studied to understand the stability of the curvature measure, and can be computed for any two nodes in a connected component of a graph.

Recalling the equations for node resistance curvature and resistance curvature, i.e. Eq. (3), it becomes clear that the main task is to understand *how the resistance distance changes in response to perturbations*. The results below from [46], are crucial for our proofs. Let $C(i, j)$ be the commute time between nodes $i, j \in V$. It is important to note that these results depend on the *normalized Laplacian*, defined in [46] as $N = D^{\frac{1}{2}} A D^{\frac{1}{2}}$, with eigenvalues $\lambda_i$, ordered such that $\lambda_1 \geq \lambda_2 \geq ....$ Here, $D$ is the diagonal matrix with inverse degrees and $A$ the adjancecy matrix. Also, as is consistent with the rest of the paper, assume our graph has $n$ nodes and $m$ edges, and $d_i$ is the degree at node $i \in V$.

**Proposition 1.** *For a graph G, let $N = D^{\frac{1}{2}} A D^{\frac{1}{2}}$ be the* normalized Laplacian *with eigen values $\lambda_1 \geq \lambda_2 \geq \cdots \geq \lambda_n$. Then, the commute time in G between nodes $i, j$ is subject to the following bounds:*

$$m\left(\frac{1}{d_s} + \frac{1}{d_t}\right) \leq C(i, j) \leq \frac{2m}{1 - \lambda_2}\left(\frac{1}{d_s} + \frac{1}{d_t}\right) \tag{10}$$

**Proposition 2.** *Consider the unweighted graph G, where each edge represents a unit resistance, i.e we consider each edge in the graph to be artificially weighted with value 1. Then the following equality holds for the commute time between nodes $i, j$:*

$$C(i, j) = 2m R_{ij} \tag{11}$$

**Proposition 3.** *If $G'$ arises from a graph G by adding a new edge, then the commute time $C'(i, j)$ between any two nodes in $G'$ is bounded by:*

$$C'(i, j) \leq \left(1 + \frac{1}{m}\right) C(i, j) \tag{12}$$

For proofs of these propositions, we refer the reader to [46]. These results create a direct connection between commute times and resistance distance, and gives insight into how commute time reacts under edge addition, and we use them directly to generate our bounds for resistance curvature.

**Theorem 4.** *If $G'$ is the graph generated by **edge addition**, then $\kappa'_R \geq \kappa_R$, with the following bound:*

$$|\kappa'_R(i, j) - \kappa_R(i, j)| \leq \frac{\Delta_{\text{add}}(d_i + d_j)}{R_{ij} - \Delta_{\text{add}}}, \tag{5}$$

*where* $\Delta_{\text{add}} := \max_{i,j \in V} \left(R_{ij} - \frac{1}{2}\left(\frac{1}{d_i+1} + \frac{1}{d_j+1}\right)\right)$.

*Proof.* Let $R'_{ij}$ be the resistance distance in $G'$. Likewise, let $C(i, j)$ be the commute distance in $G$ between nodes $i, j$ and $C'(i, j)$ be the commute time in $G'$. Then Eq. (11) and Eq. (12) ensure that $R'_{ij}$ is bounded above, by the original resistance distance in $G$:

$$2(m+1)R'_{ij} \leq 2m\left(1 + \frac{1}{m}\right)R_{ij}$$

$$R'_{ij} \leq R_{ij}$$

This follows our intuition of resistance distance very well: with the addition of an edge nodes can only get more connected. Eq. (10) also gives a nice lower bound:

$$(m+1)\left(\frac{1}{d'_i} + \frac{1}{d'_j}\right) \leq C'(i, j)$$

$$\frac{1}{2}\left(\frac{1}{d'_i} + \frac{1}{d'_j}\right) \leq R'_{ij}$$

In the case that we are adding a single edge, it is often the case that node degrees remain constant. However, the nodes that are connected by the new edge, $(i*, j*) \in E' \setminus E$, increase such that $d'_{i*} = d_{i*} + 1$ and $d'_{j*} = d_{j*} + 1$. Thus, the following lower bound holds in general for $R'_{ij}$ and we can remain agnostic to the precise location of the new edge:

$$\frac{1}{2}\Big(\frac{1}{d_i + 1} + \frac{1}{d_j + 1}\Big) \leq R'_{ij} \leq R_{ij} \tag{13}$$

And likewise, after adding $p$ edges:

$$\frac{1}{2}\Big(\frac{1}{d_i + p} + \frac{1}{d_j + p}\Big) \leq R^p_{ij} \leq R_{ij}$$

So the bounds of our *perturbed* resistance distance $R'_{ij}$ are determined by the initial network structure ($R_{ij}$) and the number connections each specific vertex has. Naturally, certain node pairs will be more strongly affected by the addition of an edge. We can define the maximum reaction to perturbation across pairs as follows:

$$\Delta_{add} := \max_{i,j \in V} \Big( R_{ij} - \frac{1}{2}\big(\frac{1}{d_i + 1} + \frac{1}{d_j + 1}\big)\Big) \tag{14}$$

This can be used to bound node resistance curvature. In an unweighted graph, we have

$$p_i = 1 - \frac{1}{2}\sum_{j \sim i} R_{ij}$$

$$p'_i = 1 - \frac{1}{2}\sum_{j \sim i} R'_{ij}$$

For $G$ and $G'$ respectively. Given that resistance can only increase, $p_i$ is clearly an lower bound for $p'_i$. Certainly an upper bound occurs when when the resistance between each one of i's neighbors maximally decreases. Thus we get the following inequality:

$$p_i \leq p'_i \leq p_i + \frac{d_i}{2}\Delta_{add} \tag{15}$$

Finally this gives the desired bound on $\kappa'_R$:

$$\kappa_R(i,j) \leq \kappa'_R(i,j) \leq \kappa_R(i,j) + \frac{\Delta_{add}(d_i + d_j)}{R_{ij} - \Delta_{add}}$$

$\square$

**Theorem 7.** *If $G'$ is the graph generated by **edge deletion**, then $\kappa'_R \leq \kappa_R$, bounded by:*

$$|\kappa'_R(i,j) - \kappa_R(i,j)| \leq \frac{1}{R_{ij} + \Delta_{del}}\Big[\frac{2}{R_{ij}}(2R_{ij} + \Delta_{del})(p_i + p_j) - \Delta_{del}(d_i + d_j)\Big],$$

*where $\Delta_{del} = \frac{2}{1-\lambda_2} - \min_{i,j \in V}(R_{ij})$ and $\lambda_2$ is the second largest eigenvalue of $N$.*

*Proof.* Now we can beg the question of how effective resistance changes when we remove an edge. By inverting our initial argument in above proof of Theorem 4, we know that after removing an edge our resistance distance can only increase. Formally, $R_{ij} \leq R'_{ij}$. For the upper bound, we can once again make an argument using Eq. (10), this time relying on the other half of the inequality. Here we need to also mention the normalized Laplacian $N'$ for $G'$, with eigenvalues $\lambda'_1 \geq \lambda'_2 \geq ... \geq \lambda'_n$.

$$C'(i,j) \leq \frac{2(m-1)}{1-\lambda_2'}\left(\frac{1}{d_i'} + \frac{1}{d_j'}\right)$$

$$R_{ij}' \leq \frac{1}{1-\lambda_2'}\left(\frac{1}{d_i'} + \frac{1}{d_j'}\right)$$

Again, we know that only the two unique vertices $(i*, j*)$ that shared an edge will have affected degrees, s.t $d_{i*}' = d_{i*} - 1$ and $d_{j*}' = d_{j*} - 1$. Moreover, from Guo et al. [28], we know that $\lambda_2 \geq \lambda_2'$. So we can loosely bound the $R_{ij}'$ as follows:

$$R_{ij} \leq R_{ij}' \leq \frac{2}{1-\lambda_2} \tag{16}$$

In fact, this applies to any number of edge deletions, as long as $G'$ stays connected. Again, we can define a maximum possible change in resistance distance across the graph:

$$\Delta_{del} = \max_{i,j \in V}\left(\frac{2}{1-\lambda_2} - R_{ij}\right) = \frac{2}{1-\lambda_2} - \min_{i,j \in V}(R_{ij}) \tag{17}$$

This leads to the following bounds on node and edge curvature, and completes the proof:

$$p_i - \frac{d_i}{2}\Delta_{del} \leq p_i' \leq p_i$$

$$(1-\lambda_2)\left[p_i + p_j - \frac{\Delta_{del}}{2}(d_i + d_j)\right] \leq \kappa_R'(i,j) \leq \kappa_R(i,j)$$

$$\kappa_R(i,j) - \frac{1}{R_{ij} + \Delta_{del}}\left[\frac{2}{R_{ij}}(2R_{ij} + \Delta_{del})(p_i + p_j) - \Delta_{del}(d_i + d_j)\right] \leq \kappa_R'(i,j) \leq \kappa_R(i,j)$$

$\square$

## B.2 Expressivity Proofs

**Theorem 5.** *Given two graphs $F = (V_F, E_F)$ and $G = (V_G, E_G)$ with scalar-valued filtration functions $f, g$, and their respective persistence diagrams $D_f, D_g$, we have $d_B(D_f, D_g) \geq \inf_{\eta: E_F \to E_G} \sup_{x \in E_F} |f(x) - g(\eta(x))|$, where $\eta$ ranges over all maps from $E_F$ to $E_G$.*

*Proof.* Considering the calculation of persistence diagrams based on scalar-valued filtrations functions, every point in the persistence diagram $D_f$ can be written as a tuple of the form $(f(e_F), f(e_F'))$, with $e_F, e_F' \in E_F$; the sample applies for $D_g$. The inner distance between such tuples that occur in the bottleneck distance calculation can thus be written as

$$\|(f(e_F), f(e_F')) - (g(e_G), g(e_G'))\|_\infty, \tag{18}$$

which we can rewrite to $\max_{C: E_F \to E_G}\{f(x) - g(C(x))\}$ for a general map $C$ induced by the bijection of the bottleneck distance. Not every map is induced by a bijection, though. Hence, if we maximise over *arbitrary* maps between the edge sets, we are guaranteed to never exceed the bottleneck distance. $\square$

## C Additional Proofs for Distinguishing Strongly-Regular Graphs

**Theorem 6** (Expressivity of curvature notions). *Both Forman–Ricci curvature and Resistance curvature* cannot *distinguish distance-regular graphs with the same intersection array, whereas Ollivier–Ricci curvature* can *distinguish the Rook and Shrikhande graphs, which are strongly-regular graphs with the same intersection array.*

*Proof.* We first show the part of the statement relating to the *Forman–Ricci curvature*. Given a distance-regular graph $G$ with $N$ vertices and intersection array $\{b_0, b_1, \ldots, b_{D-1}; c_1, c_2, \ldots, c_D\}$. Let $i, j$ be adjacent nodes in $G$. For a regular graph, we have $d_i = d_j = b_0$, where $b_0$ is a constant.

The number of triangles between two adjacent nodes $i$ and $j$ in $G$ is given by $a_1 = b_0 - b_1 - c_1$ [18]. The Forman curvature of $i, j$ is thus

$$\kappa_{\text{FR}}(i, j) := 4 - 2b_0 + 3|b_0 - b_1 - c_1|. \tag{19}$$

Given two strongly-regular graphs with the same intersection array, i.e. the same values of $b_0$, $b_1$ and $c_1$, the Forman curvature yields the same value for all pairs of adjacent nodes and cannot distinguish them. For the *resistance curvature*, the claim follows as an immediate Corollary of Theorem A [6] and described in Koolen et al. [41]. Given the resistance between two nodes depends only on the intersection array and the number of nodes in the graph, then the resistance curvature cannot distinguish two strongly-regular graphs.

The expressivity of Ollivier–Ricci curvature is strictly better, and it turns out that there are graphs with the same intersection array that we can distinguish, namely the Rook graph and the Shrikhande graph. Both graphs have the same intersection array $\{6, 3; 1, 2\}$ but differ in their first hop peripheral subgraphs [22]. It is known that 2-WL cannot distinguish these graphs. Ollivier–Ricci curvature, however, is sensitive to these differences in peripheral subgraphs with the edge curvatures for the Rook graph being: $[0.2, 0.2, 0.33, 0.33, 0.33, 0.2, 0.33, 0.33, 0.33, 0.33, 0.33, 0.33, 0.33,$ $0.33, 0.33, 0.2, 0.2, 0.33, 0.33, 0.33, 0.33, 0.33, 0.33, 0.33, 0.33, 0.33, 0.33, 0.33, 0.2, 0.33,$ $0.33, 0.33, 0.33, 0.33, 0.33, 0.33, 0.33, 0.33, 0.33, 0.33]$, and for the Shrikhande graph they are $[0, 0, 0.27, 0.27, 0.1, 0, 0.27, 0.27, 0.1, 0, 0.27, 0.1, 0.27, 0, 0.27, 0.1, 0.27, 0, 0.1, 0.27, 0.27,$ $0.1, 0.27, 0.27, 0.17, 0.17, 0.17, 0.17, 0.17, 0.17, 0.17, 0.17, 0.17, 0.17, 0.17, 0.17, 0.17, 0.17,$ $0.17, 0.17, 0.17]$, demonstrating that OR curvature can distinguish these graphs—*unlike Resistance curvature, Forman–Ricci curvature and the 2-WL test.* □

## D    Additional Stability Analysis

Given the bounds on curvature established in Section 3.1, we explore how curvature changes experimentally by analysing edge perturbations on Erdős–Rényi graphs. In particular, we provide statistics that quantify the maximal change in curvature for random graphs with varying connectivity parameters in response to edge additions and deletions. The experiment fixes the number of nodes in the ER graphs ($n = 100$), and generates a sample of 50 graphs for the selected values of $p$. For each graph in the sample, we measure the curvature $\kappa$ of all edges and calculate the standard deviation $\sigma_\kappa$ of this distribution. We then perturb the original graph by edge addition/deletion and calculate the new curvature $\kappa'$. The following tables present the *worst case* deviations in curvature, which we define as $\Delta\kappa = |\kappa - \kappa'|$, in units of $\sigma_\kappa$; in other words the maximal value of $\Delta\kappa/\sigma_\kappa$ over all sample graphs and their edges.

| Curvature | **Edge Addition:** Maximal Change ($\downarrow$) in Curvature for ER Graphs ($\Delta\kappa/\sigma_\kappa$) | | | | | | | | |
|---|---|---|---|---|---|---|---|---|---|
| | $p$=0.1 | $p$=0.2 | $p$=0.3 | $p$=0.4 | $p$=0.5 | $p$=0.6 | $p$=0.7 | $p$=0.8 | $p$=0.9 |
| $\kappa_{\text{FR}}$ | 0.582 | 0.419 | 0.334 | 0.296 | 0.253 | 0.246 | 0.232 | 0.25 | 0.315 |
| $\kappa_{\text{OR}}$ | 1.545 | 1.157 | 0.613 | 0.465 | 0.396 | 0.366 | 0.368 | 0.399 | 0.512 |
| $\kappa_{\text{R}}$ | 0.689 | 0.417 | 0.296 | 0.251 | 0.221 | 0.232 | 0.227 | 0.243 | 0.321 |

| Curvature | **Edge Deletion:** Maximal Change ($\downarrow$) in Curvature for ER Graphs ($\Delta\kappa/\sigma_\kappa$) | | | | | | | | |
|---|---|---|---|---|---|---|---|---|---|
| | $p$=0.1 | $p$=0.2 | $p$=0.3 | $p$=0.4 | $p$=0.5 | $p$=0.6 | $p$=0.7 | $p$=0.8 | $p$=0.9 |
| $\kappa_{\text{FR}}$ | 0.609 | 0.408 | 0.334 | 0.291 | 0.255 | 0.239 | 0.245 | 0.243 | 0.319 |
| $\kappa_{\text{OR}}$ | 1.397 | 1.27 | 0.623 | 0.479 | 0.394 | 0.365 | 0.347 | 0.402 | 0.482 |
| $\kappa_{\text{R}}$ | 0.75 | 0.431 | 0.336 | 0.248 | 0.229 | 0.218 | 0.225 | 0.242 | 0.312 |

## E    Additional Commentary on Counting Substructures

We find that the difference in perspective between the selected curvature notions is underscored by their respective performance when counting substructures. *Forman curvature* is an inherently local measure by definition, depending only on 3-cycles between adjacent nodes and their degrees. *Ollivier–Ricci curvature*, when used with a uniform measure, can bound the number of triangles within a locally finite graph [37] through its relation with the Watts–Strogatz clustering coefficient [67]. It

can also be shown that quadrangles and pentagons influence the OR curvature, further enhancing the expressivity of this type of curvature [37].

This is the most global perspective one can achieve using $\kappa_{OR}$ with uniform probability measures, since polygons with more than five edges do not impact the curvature valuation.

However, by changing the probability measure used by $\kappa_{OR}$, we can shift the focus towards even larger substructures. For example, the $n$th power of the transition matrix provides information on the number of $n$-paths and can therefore provide substructure information for cycles of size $n$ [43]. *Resistance curvature*, by contrast, is biased towards the largest substructures. Due to the 'global' nature of the resistance distance metric, $\kappa_R$ assigns cycles of size $\geq 5$ a positive curvature. Moreover, in a locally finite graph, one cannot use $\kappa_R$ to establish a non-trivial bound on the number of triangles (consider creating an infinite cycle between two nodes).

# F  Probability Measure for Ollivier–Ricci Curvature and Counting Substructures

The Ollivier–Ricci curvature is of particular interest because of its flexibility. While the predominant probability measure $\mu$ used by the community is *uniform* for each node, i.e. each of the node's neighbours is chosen with probability being proportional to the degree of the node. We experimented with different probability measures, one being based on expanding $\mu$ to the two-hop neighbourhood of a vertex, the other one being based on random walk probabilities. Specifically, for a node $x$ and a positive integer $m$, we calculate $\mu_{RW}$ as

$$\mu_{RW}(y) := \sum_{k \leq m} \phi_k(x, y), \tag{20}$$

with $\phi_k(x, y)$ denoting the probability of reaching node $y$ in a $k$-step random walk that starts from node $x$. Subsequently, we normalise Eq. (20) to ensure that it is a valid probability distribution. In our experiments, we set $m = 2$, meaning that at most 2-step random walks will be considered. As shown in the main paper, this formulation leads to an increase in expressivity, and we expect that further exploration of the probability measures will be a fruitful direction for the future.

We now explore to what extent the ability of the curvature to count substructures can also be improved in this way. To do this, we used powers of the transition matrix as the probability measure, as it has been shown that the $n$th power provides information on the number of $n$-paths and can therefore provide substructure information for cycles of size $n$ [43]. We find that powers of the transition matrix larger than 1 can perform better for counting the substructures, particularly for substructures larger than 3-cycles. There is also a difference between Regular and Erdős–Rényi (ER) graphs as the best transition power tends to be higher for ER graphs. We hypthosise that this may have something to do with the mixing time of the graph, as large powers should converge to the stationary distribution, and regular graphs are more 'expander-like'. The best results are obtained by taking multiple landscapes using the transition matrix powers (up to $n = 5$) and then averaging them. We show that combined with a single layer MLP, this method can perform better than using Graph Neural Network based approaches and OR curvature with the uniform measure.

| Method | Counting Substructures (MAE ↓) | | | |
| --- | --- | --- | --- | --- |
| | Triangle | Tailed Tri. | Star | 4-Cycle |
| GCN | 0.4186 | 0.3248 | **0.1798** | 0.2822 |
| $\kappa_{OR}$ Filtration | 0.2321 | 0.2395 | 0.3393 | 0.3089 |
| $\kappa_{OR}$ Filtration with transition matrix powers | **0.1956** | **0.2095** | 0.3212 | **0.2680** |

| Method | Optimal Transition Power | |
| --- | --- | --- |
| | ER | Regular |
| Triangle | 2 | 1 |
| Tailed Triangle | 4 | 3 |
| Star | 4 | 2 |
| Chordal Cycle | 2 | 2 |
| 4-Cycle | 8 | 3 |

*Table 5:* Computation time in seconds for discrete curvature on varying Erdős–Rényi graph sizes with $p = 0.3$.

| Number of Nodes | $\kappa_{FR}$ | $\kappa_{OR}$ | $\kappa_R$ |
|---|---|---|---|
| 10 | 0.000 | 0.002 | 0.020 |
| 50 | 0.001 | 0.038 | 0.700 |
| 100 | 0.005 | 0.247 | 6.610 |
| 250 | 0.054 | 4.720 | 252.850 |
| 500 | 0.380 | 59.270 | 6414.970 |
| 1000 | 2.920 | 1040.700 | 74 366.070 |

*Table 6:* Computation time for different numbers of Erdős-–Rényi graphs in a reference set ($n = 10$ and $p = 0.3$) with different distribution distance measures

| Number of Graphs | Degree + MMD | | Orbit + MMD | | Curvature + MMD | | Curvature + Landscapes | |
|---|---|---|---|---|---|---|---|---|
| 20 | 3.6 | ms | 217.0 | ms | 4.2 | ms | 12.5 | ms |
| 50 | 10.9 | ms | 459.0 | ms | 12.4 | ms | 20.9 | ms |
| 100 | 34.1 | ms | 887.0 | ms | 37.9 | ms | 34.4 | ms |
| 200 | 120.0 | ms | 1960.0 | ms | 133.0 | ms | 80.4 | ms |
| 500 | 678.0 | ms | 6740.0 | ms | 727.0 | ms | 144.0 | ms |
| 1000 | 2620.0 | ms | 19 900.0 | ms | 2680.0 | ms | 359.0 | ms |

## G  Computational Complexity

Persistence diagrams of 1-dimensional simplicial complexes, i.e. graphs, can be computed in $\mathcal{O}(m \log m)$ time where $m$ denotes the number of edges. Empirically, when calculating different curvature measures for different sizes of graphs, we find that Forman curvature scales well to large graphs, whereas OR and resistance curvatures can be used for smaller graphs and in cases that require a more expressive measure. Note that there are significantly faster ways to calculate resistance curvature as an approximation [65]. A majority of works on GGMs focus on small molecule generation, where any of these curvatures can be used with minimal pre-computation. Table 5 depicts the computational complexity of various curvature calculations on Erdős–Rényi graphs whilst Table 6 and Table 7 compares the complexity to methods based on MMD. We find that calculating persistence diagrams, turning these to persistence landscapes, averaging these and then calculating a distance takes a similar amount of time compared to MMD for different sizes of graphs and for different numbers of graphs in the reference set. Interestingly, our approach scales better than MMD as both the number of graphs in the reference set increases and when the size of the graphs increases. This will be important for comparing distributions of large data sets such as the commonly used Zinc dataset or QM9. Overall, we find that our method can be easily applied in practical use cases, especially given that models for graph generation tyically generate graphs with well under 1000 nodes.

*Table 7:* Computation time for a fixed number of Erdős–Rényi graphs in a reference set with different sizes ($p = 0.3$) with different distribution distance measures

| Number of Graphs | Curvature + MMD | | Curvature + Landscapes | |
|---|---|---|---|---|
| 10 | 2.3 | ms | 9.0 | ms |
| 20 | 4.5 | ms | 12.5 | ms |
| 50 | 15.8 | ms | 20.9 | ms |
| 100 | 93.6 | ms | 34.4 | ms |
| 200 | 556.0 | ms | 80.4 | ms |
| 500 | 727.0 | ms | 144.0 | ms |
| 1000 | 2800.0 | ms | 364.0 | ms |

# H Ethical Concerns

We have proposed a general framework for comparing graph distributions focusing primarily on method and theoretical development rather than on potential applications. We currently view drug discovery as being one of the main application areas, where further experiments may be required, but we have no evidence that our method enhances biases or causes harm in any way.

