# OpenReview forum: "Curvature Filtrations for Graph Generative Model Evaluation"
_NeurIPS.cc/2023/Conference — NeurIPS 2023 poster_

### Official Review · Reviewer_YxdW · 2023-06-23

**Soundness:** 3 good
**Presentation:** 3 good
**Contribution:** 3 good
**Rating:** 6
**Confidence:** 4

**Summary:**

In this manuscript, the authors proposed a special curvature-related graph descriptors for the evaluation of graph generative model. The key idea is to evaluate the discrete curvatures on graphs and construct a curvature-based filtration process. The corresponding persistent Landscape based metric is used for the evaluation of graphs. This metric is robust, stable, and expressive.

**Strengths:**

A unique curvature-based TDA model is developed and used for the first time for the evaluation of graph generative model. The developed curvature-based metric is robust, stable, and expressive.

**Weaknesses:**

The advantage to use curvature as filtration parameter is not clear. The advantage of the special filtration process and final vectorization approach is not clear.

**Questions:**

1)The construction of curvature-based TDA model is unique. However, its advantages are not clear. Once we have a graph, there are various way to define filtration parameters (or functions), what is the advantage of using curvature as the filtration parameter? Further, persistent curvature models have been developed and used in various molecular classification and property prediction. The authors are suggested to add some discussions and comparisons.

JunJie Wee and Kelin Xia, "Forman persistent Ricci curvature (FPRC) based machine learning models for protein–ligand binding affinity prediction." Briefings In Bioinformatics, 22 (6), bbab136 (2021)

JunJie Wee and Kelin Xia, "Ollivier persistent Ricci curvature-based machine learning for protein-ligand binding affinity prediction." Journal of Chemical Information and Modeling, 61 (4), 1617-1626 (2021)

2) The filtration process is kind of weird. Note that positive Forman/Olivier Ricci curvatures are usually associated with certain community or cluster regions, while negative Ricci curvature are associated with the link or bridge regions. A filtration process from the largest Ricci curvature to smallest curvature will tend to reveal these cluster-type of information and seems to be more reasonable. But in this paper, the authors use the opposite way of going from the smallest to largest. Can the authors add some more discussions and explanations?

3) In the sublevel set filtration, the final simplicial complex is just the original graph. In this way, all Betti_1 cycles will never been killed. That is all Betti_1 death values are infinity and no important death information will be obtained for Betti_1. However, if the authors use a general filtration process to generate simplicial complexes (with dimension 2 or above), 2-simplexes can be generated and Betti_1 bars can provide more information. Can the authors add some more discussions and explanations or comparison?

4)There are many different vectorization models for TDA. Why the authors choose to use persistent landscape? What is the advantage of persistent landscape for this model?

5) Forman-Ricci curvature is a combinatorial model designed for cell complexes. It is slightly “unfair” to directly compare its performance on graph with Ollivier Ricci curvature, which is designed for graphs. It may be better to add some more explanations about the background of Forman-Ricci curvature to avoid the confusion from readers not from this area!


**Limitations:**

More discussions and explanations are needed to fully demonstrate the advantage of their model.

---

> ### Author Rebuttal · Authors · 2023-08-04
>
> Thanks for the positive assessment of our work! We will include the suggested citations and are very happy to provide additional clarifications!
>
> > The advantage to use curvature as a filtration parameter is not clear. The advantage of the special filtration process and final vectorization approach is not clear.
>
> We summarize main points; see long response below:
>
> 1. We use curvature because it provides us with advantageous stability and expressivity properties (Sections 3.1 and 3.2).
> 2. Moreover, curvature turns out to be a characteristic property of many geometric graphs such as molecules (we are very grateful for the additional literature you mentioned, which also serves to underscore exactly this point!)
> 3. Finally, the vectorization process of persistence landscapes provides us with unique averages that are (a) easy to compute, and (b) permit us to compare distributions of graphs, while benefiting from the overall stability guarantees of TDA.
>
> > The construction of curvature-based TDA model is unique. However, its advantages are not clear. Once we have a graph, there are various ways to define filtration parameters (or functions), what is the advantage of using curvature as the filtration parameter?
>
> **Thanks for highlighting the unique aspect of our method**! We observe two advantages of curvature-based TDA:
>
> - _Stability_: Thm 1 implies that the stability of our evaluation method depends on the stability of the filtration functions we choose. We understand curvature’s stability with respect to graph perturbations and show bounds in Thms 2,3,4. This shows that our method is robust to perturbing distributions of graphs (i.e., bounded perturbations of graphs lead to bounded perturbation of curvature, which leads to bounded perturbation on landscape, which ultimately leads to a bounded perturbation on the average landscape).
>
> - _Expressivity_: Curvature itself is an expressive graph descriptor, which we motivate by analyzing its ability to distinguish strongly regular graphs and count substructures. To develop a stable/practical method for evaluating GGMs we want to use the most expressive graph descriptors that remain computationally viable.
>
> **Although other filtrations can be used in our framework, we are arguing that the combination of curvature’s expressivity, stability, and computability makes it the best choice for evaluating GGMs. We will emphasise this aspect in a revision.**
>
> >  Further, persistent curvature models have been developed and used in various molecular classification and property prediction. [...]
>
> Thanks so much for these suggestions! We apologize for missing these relevant references for using curvature in the _classification_ of geometric graphs. **We will include them and discuss them as related work.**
>
> > The filtration process is kind of weird. Note that positive Forman/Olivier Ricci curvatures are usually associated with certain community or cluster regions, while negative Ricci curvature are associated with the link or bridge regions.[...] But in this paper, the authors use the opposite way of going from the smallest to largest. Can the authors add some more discussions and explanations.
>
> That's a great point, thanks for raising it! Our framework works for both filtration directions, and we initially wanted to capture cycles 'earlier' (i.e. for smaller values of curvature). The reviewer's suggestion is great, and we will **highlight the possibility of a different filtration direction**, though our preliminary experiments do not indicate it makes a difference.
>
> > [...] In this way, all Betti_1 cycles will never been killed. That is all Betti_1 death values are infinity and no important death information will be obtained for Betti_1. However, if the authors use a general filtration process to generate simplicial complexes (with dimension 2 or above), 2-simplexes can be generated and Betti_1 bars can provide more information. [...]
>
> Thanks for this suggestion! It is possible to move to higher-dimensional simplices, but we may lose the efficiency of computing PH using graph algorithms. What we could do to address this problem is consider [extended persistence](https://link.springer.com/article/10.1007/s10208-008-9027-z), which we would like to explore in a future work. When including 2-simplices and re-running the classification experiments between strongly-regular graphs, we find that the numbers in Table 1 changes as follows:
>
> | Model | sr16622 | sr261034 | sr281264 | sr401224 |
> | --- | --- | --- | --- | --- |
> | Without $2$-simplices   | 1.00     | 0.89     | 1.00    | 0.93       |
> | With $2$-simplices          | 1.0           | 0.98           | 1.0           | 0.94            |
>
> Our framework supports higher-dimensional information, but given the computational considerations, **we will add a note in a revision of the paper.** Thanks for the suggestion!
>
> > [...] What is the advantage of persistent landscape for this model?
>
> We find that lifting the calculation to a Banach space has useful properties, such as providing us with unique averages and ways of performing statistical testing. Moreover, persistence landscapes can be calculated quickly and afford efficient distance calculations. **We will emphasize these points more in our revision.**
>
> > Forman-Ricci curvature is a combinatorial model designed for cell complexes. [...] It may be better to add some more explanations about the background of Forman-Ricci curvature to avoid confusion from readers not from this area.
>
> Thanks for the suggestion! Forman-Ricci curvature seems to be rather know in ML, e.g. in the context of graph neural networks. **We will discuss more related work on [Forman--Ricci curvature](https://openreview.net/forum?id=7UmjRGzp-A) in a revision.**
>
> Again, thank you for your feedback on our manuscript. If you have any additional questions, please let us know! Should you be satisfied with our responses to your queries, we would be grateful if you considered reflecting this in your score.

---

> > ### Comment · Reviewer_YxdW · 2023-08-11
> >
> > All my questions are well addressed. I have no further comments.

---

> > > ### Author Response · Authors · 2023-08-16
> > >
> > > Thank again for your feedback. Your recognition of the **strengths of our work is very valuable!** Please consider adjusting your rating in light of our comments being satisfactorily. Since our submission received only three reviews, we would appreciate any additional support that provides us with the opportunity to present our work at NeurIPS.

---

### Official Review · Reviewer_4GUL · 2023-06-28

**Soundness:** 3 good
**Presentation:** 3 good
**Contribution:** 3 good
**Rating:** 7
**Confidence:** 3

**Summary:**

The paper proposes a new set of graph features: first, 3 variants of graph curvature are calculated. Second, persistence landscapes are calculated over edge-based filtrations via curvature. Third, sets of graphs are compared via comparing their landscapes.
Theoretical properties of landscapes (stability, expressivity) are studied. Experiments for both synthetic and real graph sets are provided.

**Strengths:**

The research is original, afaik, since only few papers are dedicated to the developpement of new *measures* to evaluate graph generative models. However, the problem itself is an important one, having applications in molecular design.
The manuscript presents new theoretical results: stability w.r.t. changes in graph structure and expressivity of landscapes vs. raw filtrations (the results concern graph curvature).
Experiments are quite diverse.
The paper is well written and easy to follow.

**Weaknesses:**

1. While the paper provides versatile approaches to comparison of graph datasets, the single end-to-end algorithm to evaluation of graph generative models and corresponding score is not formulated. I expected to find something like FID [1] for graphs. But in the end, you have a set of persistence landscapes, not one number. These landscapes should be aggregated somehow?
2. No experiments with real graph generative models.

[1] Heusel, M., Ramsauer, H., Unterthiner, T., Nessler, B., & Hochreiter, S. (2017). Gans trained by a two time-scale update rule converge to a local nash equilibrium. Advances in neural information processing systems, 30.

**Questions:**

1. I am surprised that you have no experiments with a real graph generative model, instead you only compare manually prepared sets of graphs. The manuscript will benefit from such experiments.
2. I don't understand why you don't use MMD in Section 4.5, you can calculate pairwise kernels for two sets of persistence landscapes.
3. What is the dimension of persistence homology you calculate? How do you aggregate landscapes for different dimensions ?
4. Is your approach is sensitive to outliers (outlier graphs in datasets) ?
5. Why you have choosed edge-based filtration instead of node-based (like node degree)? Is it more expressive?

**Limitations:**

Authors adequately addressed the limitations and potential negative societal impact of their work.

---

> ### Author Rebuttal · Authors · 2023-08-04
>
> Thank you for your positive assessment of our work! We are looking forward to clarifying any additional questions you might have!
>
> > While the paper provides versatile approaches to comparison of graph datasets, the single end-to-end algorithm to evaluate graph generative models and corresponding score is not formulated. I expected to find something like FID [1] for graphs. But in the end, you have a set of persistence landscapes, not one number. These landscapes should be aggregated somehow?
>
> We aggregate the set of landscapes into an average landscape, and compare different averages via their $p$-norms (this is in fact one of the advantages of the persistence landscape formulation: it affords an easy-to-calculate mean). Similar to FID, our score is a difference between the representative landscapes for original and generated distributions of graphs, whose confidence can be evaluated by a $t$-test, for instance. **We will emphasise these aspects in a revision.**
>
> > I am surprised that you have no experiments with a real graph generative model, instead you only compare manually prepared sets of graphs. The manuscript will benefit from such experiments.
>
> This is a great suggestion! For now, we chose not to compare outputs from real graph generative models due to the fact there is no ground truth in this case; the score between a given model’s generated and reference set doesn’t necessarily tell you how good that score is. We agree that the measure should be used to evaluate these models in the future but we first wanted to establish the new measure in a principled fashion and show how it overcomes the limitations of current approaches.
>
> > I don't understand why you don't use MMD in Section 4.5, you can calculate pairwise kernels for two sets of persistence landscapes.
>
> Thanks for this suggestion! You are absolutely right to point out the existence of such kernels. However, due to the criticism of MMD outlined by [O’Bray et al.](https://openreview.net/forum?id=tBtoZYKd9n) (i.e. they find the final score is very sensitive to parameter choices and it has poor scalability), we chose to provide a method for the community which does not rely on it. Therefore, we have outlined a method that only requires defining the choice of curvature and has improved scalability. Nevertheless, our framework would potentially allow for replacing the distance calculations with a kernel. **We will emphasise this aspect in a revision.**
>
> > What is the dimension of persistence homology you calculate? How do you aggregate landscapes for different dimensions?
>
> For now, we calculate $0$-dimensional and $1$-dimensional features (connected components and cycles) over the filtration, which can be computed efficiently using existing graph algorithms. However, our method also would permit higher-order topological features (at the expense of scalability). Our aggregation procedure then follows these steps:
>
> - We generate a landscape for each graph $G$, which can be decomposed into persistence landscape functions $\lambda_{G,0}$ and $\lambda_{G,1}$. These are _piecewise linear_ functions that encode the $0$- and $1$-dimensional persistent homology, respectively.
> -  We then aggregate the functions separately into average landscape functions $a_0 := \lambda_{\mathrm{avg}, 0}$ and $a_1 := \lambda_{\mathrm{avg}, 1}$ for the whole distribution.
> - Since persistence landscapes come equipped with ($p$-)norms, we use the supremum norm in our experiments. Our final summary of a distribution results in a two dimensional vector $V[\|\|a_0\|\|_\infty, \|\|a_1\|\|_\infty]$.
> - Finally, to compare the two distributions, we compare the Euclidean distance between these vectors.
>
> **We will highlight this calculation better in a revision.**
>
> > Is your approach is sensitive to outliers (outlier graphs in datasets) ?
>
> Thanks for raising this point! We believe that, due to the properties of persistence landscapes, we can do statistical testing on a distribution of graphs to detect outliers in the distributional sense (since such outlier graphs would result in a highly different set of topological descriptors).  Moreover, our ability to detect outliers is underscored by the expressivity of the curvature filtrations. **We will highlight this aspect better in the revision of the paper, potentially also making a connection to the [stability theorem(s)](https://arxiv.org/abs/2006.16824) in persistent homology, which provides conditions under which persistence diagrams (and, subsequently, also persistence landscapes) remain stable.**
>
> > Why you have chosen edge-based filtration instead of node-based (like node degree)? Is it more expressive?
>
> We choose to do an edge-based filtration over a node-based filtration, due to the fact that curvature is naturally defined over edges. If we were to make it node-based (e.g. mean of curvature of incoming edges) then indeed we would lose some information and be less expressive; preliminary experiments on strongly-regular graphs appear to be in support of this hypothesis. **We will add a brief discussion about this point.**
>
> Again, thank you for your feedback on our manuscript. If you have any additional questions, please let us know! Should you be satisfied with our responses to your queries, we would be grateful if you considered reflecting this in your score.

---

> > ### Comment · Area_Chair_eMKo · 2023-08-19
> > **Thanks for the response**
> >
> > Dear authors,
> >
> > Thanks for your efforts to clarify the concerns raised by the reviewer. Although the reviewer has not responded yet, I'd like to assure you that your response will be incorporated into the reviewers' discussion and my final recommendation.
> >
> > Best wishes,
> > AC

---

> > > ### Author Response · Authors · 2023-08-19
> > >
> > > Thank you very much, we appreciate your support as well as the efforts by all reviewers.

---

> > ### Comment · Reviewer_4GUL · 2023-08-19
> > **Response**
> >
> > Thank you for the detailed response, my questions are addressed. I hope that your manuscript will benefit from clarifications, especially by adding more details about calculating your "graph distribution distance". I prefer to leave my score unchanged (already positive).

---

### Official Review · Reviewer_juxy · 2023-07-06

**Soundness:** 2 fair
**Presentation:** 2 fair
**Contribution:** 2 fair
**Rating:** 5
**Confidence:** 3

**Summary:**

This paper proposed a TDA-based graph generation model evaluation method using the discrete Ricci curvature of the graph as a filtration function. The proposed comparison between the two graphs is theoretically shown to have good properties for graph analysis, and the comparison is verified in experiments.

**Strengths:**

A new way (distance) of comparing  two graphs from a new perspective is proposed, and it is theoretically shown that the properties required for graph comparison are satisfied.

**Weaknesses:**

- The main concern is the consistency between the issues in the paper and the proposed methodology and validation. Although the issue of the paper is supposed to be a method for evaluating graph generation models, the proposed method is about a measure of the distance between two graphs. The two are related, of course, but there is insufficient mention of comparison methods between graph-generating models and comparison with other comparison methods. If the issue is the importance of defining good distances, the paper should be constructed as such, and if the issue is the evaluation of graph generation models, there should be a full discussion of that.

- In any case, the key to the proposed method is the definition of the distance between two graphs, but most of the comparisons are in the Ricci curvature and TDA domains only, and the distance and similarity between graphs in other domains are not discussed. The following are representative of the distances and similarities between graphs. It is unnatural in light of the scope of NeurIPS that these are not mentioned.
	- Graph edit distance
	- Y. Bai et. al, SimGNN: A Neural Network Approach to Fast Graph Similarity Computation, WSDM '19

- This is a non-essential question (with little impact on the evaluation), but why do you use Landscape for comparisons between persisntent diagrams? Many have been proposed such as Wesserstein distance, but is there a reason why Landscape is the best? Or is this an issue for future study?

**Questions:**

Please answer the Weakness question above.

**Limitations:**

The authors do not explicitly address Limitation. On the other hand, the study is oriented toward reducing the social impact on the limitations of conventional methods and does not promote adverse effects.

---

> ### Author Rebuttal · Authors · 2023-08-04
>
> We thank the reviewer for taking time in their review and providing valuable feedback. We have tried to address the weaknesses that you have highlighted. We believe most of the comments arise from the confusion between traditional distances between *graphs* and a distance between *distributions of graphs* (the main novelty of our work). We will better articulate this difference in the revised version of our paper.
>
> > Although the issue of the paper is supposed to be a method for evaluating graph generation models, the proposed method is about a measure of the distance between two graphs.
>
> We apologise for this misunderstanding: in the paper, we are in fact proposing a distance between *distributions* of graphs, which enables us to evaluate graph generation models, where there is a reference set of graphs and a generated set. You are right in pointing out that there is a connection to distances between *individual* graphs. We use an example of distinguishing pairs of graphs in our work in order to show the **expressivity** of our measure (and to relate to the vaster literature on graph comparison) but we actually generate a distance between two sets of graphs (in this case, only one graph per set). This is in contrast to graph kernels and the graph edit distance which can not measure a distance between graph distributions. **We will rewrite this in a revision and clarify this point.**
>
> > There is insufficient mention of comparison methods between graph-generating models and comparison with other comparison methods
>
> Current comparison methods for evaluating graph generation use descriptor functions (degree, Laplacian, clustering coefficient) with MMD and we have conducted experiments to show that our approach is more stable (experiment 4.2), is better at distinguishing graph datasets and thus more expressive (experiment 4.5), and scales better (appendix F). We have also highlighted other weaknesses of previous methods such as not being robust to parameter choices, an issue alleviated by our method. **We will highlight these aspects in a revision; please let us know if there are other specific comparisons you would like us to conduct.**
>
> > [...] The key to the proposed method is the definition of the distance between two graphs, but most of the comparisons are in the Ricci curvature and TDA domains only, and the distance and similarity between graphs in other domains are not discussed.
>
> As mentioned above, the distance we describe is not just a distance *between graphs* but a distance between two **distributions of graphs**. The choice of Ricci curvature combined with TDA is specifically done to allow this comparison between two sets of graphs and not just between two graphs (which cannot be used to evaluate graph generation). **We will discuss methods for computing distances between graphs as related work in a revision of the paper.**
>
> > This is a non-essential question (with little impact on the evaluation), but why do you use Landscape for comparisons between persistent diagrams? Many have been proposed such as Wesserstein distance, but is there a reason why Landscape is the best? Or is this an issue for future study?
>
> We selected persistence Landscapes because of their favorable theoretical and empirical properties. Persistent Landscapes allow us to calculate distances between two *distributions* of graphs and not just between two graphs. This is due to the fact that, unlike Wasserstein distances, persistence landscapes allow us to do distributional analysis (whereas if working with persistence diagrams, their [means are non-unique and hard to calculate](https://arxiv.org/abs/1206.2790)). This means we can get average topological descriptors for all graphs in a distribution and then get a distance to the average of another distribution. There are indeed other functional summaries that could be used and we could also use a deep learning-based evaluation. **Thanks for raising this point; we will discuss these ideas in the revision and consider them for future work.**
>
> Thanks for your assessment, please let us know about any other questions you might have! Should you be satisfied with our responses to your concerns, we would be grateful if you considered raising your score accordingly.

---

> > ### Comment · Reviewer_juxy · 2023-08-12
> > **Thank you for your clarification.**
> >
> > Thank you for your clarification.
> > One point I am still not clear on regarding the first and third answers is that Section 3 says to use a permutation test like method for the distance of the distribution. This seems to be computable given the distance between the two graphs. Is the introduction of permutation test, etc. a contribution, or does a method using Ricci curvature have to be used to use permutation test, etc.? Or, does the method using Ricci curvature have good features for calculating permutation test, etc. (other distances have disadvantages)? Please clarify these matters.
> > I received a clear answer regarding the second and fourth.  I cannot confirm the revised version due to the rules at this time, but I hope it will be revised so that the entire required content can be captured in the main body.

---

> > > ### Author Response · Authors · 2023-08-12
> > > **Clarification of permutation test experiment**
> > >
> > > Thank you for your response. For the permutation test experiment, we have 10 graphs coming from two distributions of graphs and the goal is to quantify if these two distributions are different. This is related to graph generative model evaluation where we want to understand if our generated graphs match the training distribution. Our method, using a Ricci Curvature filtration with TDA,  means we can get a distance between these two distributions of graphs (persistence landscapes allow us to do distributional analysis). Note that we can use any filtration function here. This is in contrast to methods which compare distances between two graphs such as the graph edit distance. The purpose of the permutation test is to then understand if this distance between distributions is significant (ie. does a distance of 3 between the two distributions mean the distributions are significantly different). We permute graphs from both samples, randomly selecting sets of equal size, measure their respective distances, and finally aggregate the fraction of distances which are higher than the original. This analysis (permutation/two-sample testing) is only possible because we provide a distance between graph distributions, unlike a majority of kernel-based approaches, which provide a distance between all individual graphs.
> > > We find that our method is expressive enough to distinguish these distributions and performs better than previous approaches which use descriptor functions + MMD.
> > > I hope we have been able to clarify but if you have any additional questions, please let us know

---

> > > > ### Comment · Reviewer_juxy · 2023-08-13
> > > > **Thank you for your reply.**
> > > >
> > > > Thank you for your reply.
> > > > There seems to be a misunderstanding either in my understanding of the paper or the question. My understanding is that filtration applies to a single graph structure data. If it applies to a set (distribution) of graphs, please clarify where that statement is found. If it is correct to apply it to a single graph, then it is not due to the use of curvature filtration that we are able to compare sets from the beginning of Section 3 (1.-3. of Our Method), but rather due to the third permutation test. The reason for this is not due to the use of curvature filtration. (It seems to be stated as such in Section 3.) If it is by the third definition in Section3 that the distance between distributions can be calculated, then I think one of the following logics is required to accept that the contribution of this paper can be used to calculate the distance between distributions.
> > > > - The introduction of the third definition in Section3 is one of the contributions of this paper.
> > > > - Other methods of calculating the distance between graphs do not apply the third definition in Section 3, or cause problems when they do.
> > > > I would like clarification on the above points.

---

> > > > > ### Author Response · Authors · 2023-08-13
> > > > > **Clarification on distributional analysis and thank you for engaging**
> > > > >
> > > > > You are correct that the filtration is applied per graph and from this we get a persistence landscape per graph. But persistent landscapes allow for easy calculation of averages so we can get an average landscape for all graphs in a set. We can do this for different sets of graphs and compare their average landscapes via the *p*-norm. This is a distance between *distributions* of graphs. Note that we can average over *any* number of graphs in the set. Therefore it is due to the curvature filtration **and** the use of persistence landscapes that we can get distances between sets of graphs. This is in contrast to methods which compare distances between two graphs such as graph edit distance and graph kernels that cannot be used for this task. The permutation test is only used to see if this distance is *significant*.
> > > > >
> > > > > - The permutation test isn't required to measure a distance between distributions. Only to get a level of significance of the distance value given from our method.
> > > > >
> > > > > - Other kernel-based methods do not measure a distance between distributions and so the permutation test experiment does not work in this case.

---

> > > > > > ### Comment · Reviewer_juxy · 2023-08-14
> > > > > > **Re:**
> > > > > >
> > > > > > I understood your claim to be that computing the average of a landscape over a set of graphs allows for computation over a distribution. I don't know what the landscape average represents, but I assume it extracts representative features of no distribution (which is not obvious). This would be analogous to using the center or center of gravity of the distribution, which could be computed using other graph distances. In addition, it seems a wild argument to express the characteristics of the distribution only in terms of the mean. In the case of a distribution discussion, the degree of variability of the data is also important. Just as even a normal distribution requires a mean and variance. I don't believe the landscape average includes the distribution shape, so please let me know if the proposed method can obtain the distribution shape.  In any case, I think it is a leap to claim that curvature filtration made it possible to calculate the distance of the distribution. On the other hand, if the argument is that curvature filtration is a good distance definition for that purpose, it makes sense.

---

> > > > > > > ### Author Response · Authors · 2023-08-14
> > > > > > >
> > > > > > > > This would be analogous to using the center or center of gravity of the distribution, which could be computed using other graph distances.
> > > > > > >
> > > > > > > Could you clarify how averages can be taken using graph distances? We understand how you could take one graph that minimised the average distance but the difficulty in averaging graphs is why previous methods use descriptor functions + MMD and why we use curvature filtrations + persistence landscapes.
> > > > > > >
> > > > > > > > In the case of a distribution discussion, the degree of variability of the data is also important. Just as even a normal distribution requires a mean and variance. I don't believe the landscape average includes the distribution shape, so please let me know if the proposed method can obtain the distribution shape.
> > > > > > >
> > > > > > > A benefit of using persistence landscapes in our method is that we can take variances and do distributional analysis. We take distances between means of the distributions, as do previous methods which use descriptor functions + MMD. Future work could also consider higher order moments of the distribution.
> > > > > > >
> > > > > > > > I don't know what the landscape average represents, but I assume it extracts representative features of no distribution.
> > > > > > >
> > > > > > > The average landscape collects and aggregates the structural features of the graphs in the distribution. The ability to detect and precisely describe the structural features of a graph comes from the choice of filtration: we choose curvature and back up this decision with our expressivity experiments.
> > > > > > >
> > > > > > > > In any case, I think it is a leap to claim that curvature filtration made it possible to calculate the distance of the distribution. On the other hand, if the argument is that curvature filtration is a good distance definition for that purpose, it makes sense.
> > > > > > >
> > > > > > > Could you clarify where this claim is made so that we can make it more clear in the paper? We claim that using persistent landscapes allow us to compute distances between distributions and that curvature is a *good* (expressive and stable) choice of filtration, and go on to show that the distance yielded by our method is more expressive, stable, and scales better than previous approaches.
> > > > > > >
> > > > > > > > Although the issue of the paper is supposed to be a method for evaluating graph generation models, the proposed method is about a measure of the distance between two graphs.
> > > > > > >
> > > > > > > We hope that we have been able to clarify the main weakness that was initially given. We can get distances between distributions which allows us to evaluate graph generative models. We have shown the benefits of our method over previous approaches which use descriptor functions + MMD.

---

> > > > > > > > ### Comment · Reviewer_juxy · 2023-08-14
> > > > > > > > **Re:**
> > > > > > > >
> > > > > > > > Apparently, the discussion is not engaging. I would like to confirm that your claim is one of the following:
> > > > > > > > - The contribution is to be able to calculate the distance between the distributions of a graph. In other words, it was built for the first time.
> > > > > > > > - If the distance between graphs can be defined, the distance between distributions can be computed (with or without good performance), but curvature filtration performs much better than conventional distances.
> > > > > > > > - If the distance between graphs can be defined, the distance between distributions can be computed (with or without good performance), but distribution calculations with curvature filtration and landscape achieve much better performance than previously possible.
> > > > > > > >
> > > > > > > > At the time I first read the paper, I thought your argument was the second, as there was much discussion of the distance between the two graphs. However, since rebuttal I thought you were making the first claim because you have emphasized that the distance between distributions can be computed without admitting that you can compute the distance between distributions with the distance between other graphs. It was my opinion that other distances can formally calculate distances between distributions (e.g., distances between balance point) and that comparisons should be made even if they do not provide sufficient distributional information, but you seem to have been arguing whether those methods make sense or not. Maybe it's because I didn't say enough about this. From the comments I have received, it appears that your claim is closer to the third one.
> > > > > > > > It is difficult to accept the assumption that there is nothing that gives a meaningful distance between graph distributions without any evidence. This is a check to determine if this is a critical issue for this paper.
> > > > > > > >  It is unclear how the landscape averages are calculated, but how different is it from taking the average of the landscapes and using the landscapes to take the distance matrix within the distribution and taking their center or balance point? I think it would be appropriate at the feature stage. In any case, I presume that you are constructing one representative graph.
> > > > > > > >  Does the above comment mean that it is possible to calculate the equivalent of variance because it can be calculated at the feature stage? Or does it mean that the average also contains information equivalent to variance? (although I don't believe the latter) In any case, I think these details are clearly unclear within the paper.
> > > > > > > >  Another way of thinking about it might be to provide graph features (landscapes) as a framework for users to compare distributions as they wish, without limiting the method of comparison between distributions. In this case, the phrase "for evaluating graph generative models" would be misleading.

---

> > > > > > > > > ### Author Response · Authors · 2023-08-15
> > > > > > > > >
> > > > > > > > > We thank the reviewer for the detailed discussion, and we hope to
> > > > > > > > > provide additional information:
> > > > > > > > >
> > > > > > > > > **In a nutshell:** Using persistence landscapes, we are essentially
> > > > > > > > > obtaining a space (Banach space) that permits us to do averaging/distributional
> > > > > > > > > analysis. Ideally, we would average directly over the space of graphs
> > > > > > > > > (adjacency matrices) but **averages are not well-defined in this space**. We
> > > > > > > > > try and retain as much of the essential graph information as possible
> > > > > > > > > using curvature as the filtration and have arguments showing the
> > > > > > > > > expressivity of this measure.
> > > > > > > > >
> > > > > > > > > Our claim is the following:
> > > > > > > > >
> > > > > > > > > - The contribution is *an improvement* in calculating the distance
> > > > > > > > >   between graph distributions. This is typically done using descriptor
> > > > > > > > >   functions, coupled with MMD [1] and we show that **our method is more expressive,
> > > > > > > > >   stable and scales better**.
> > > > > > > > >
> > > > > > > > > > If the distance between graphs can be defined, the distance between
> > > > > > > > > > distributions can be computed
> > > > > > > > >
> > > > > > > > > To our knowledge, there is no previous work that used a direct distance
> > > > > > > > > between graphs, such as the graph edit distance, to get a distance between distributions
> > > > > > > > > of graphs. **Please let us know if you are aware of such previous
> > > > > > > > > work.**
> > > > > > > > >
> > > > > > > > > We do not use this as a baseline because we are not sure how to go
> > > > > > > > > pairwise graph distances to a proper distributional distance; we believe
> > > > > > > > > that this would **constitute a novel approach in itself.**
> > > > > > > > >
> > > > > > > > > Following your comment, could you kindly clarify how, once you have the
> > > > > > > > > balance point (such as the graph that minimises the average pairwise
> > > > > > > > > distance to all other graphs), you can take an average or interpolate
> > > > > > > > > between the graphs in your set? **For instance, how would you calculate
> > > > > > > > > the average of two graphs which are edit distance X away from
> > > > > > > > > each other?**
> > > > > > > > >
> > > > > > > > > We believe that it is possible to calculate the distance between
> > > > > > > > > distributions of graphs by using feature vectors as proxy information,
> > > > > > > > > but it is precisely this approach that was criticised (see [1] for more
> > > > > > > > > details).
> > > > > > > > >
> > > > > > > > > > It is unclear how the landscape averages are calculated, but how
> > > > > > > > > > different is it from taking the average of the landscapes and using
> > > > > > > > > > the landscapes to take the distance matrix within the distribution and
> > > > > > > > > > taking their center or balance point?
> > > > > > > > >
> > > > > > > > > We think these approaches should be equivalent. But notice that the
> > > > > > > > > average landscape (the landscape at the centre of mass), i.e. our
> > > > > > > > > approach, is **well-defined**, whereas the centre of mass of adjacency
> > > > > > > > > matrices using the edit distance is **not**.
> > > > > > > > >
> > > > > > > > > > Does the above comment mean that it is possible to calculate the
> > > > > > > > > > equivalent of variance because it can be calculated at the feature
> > > > > > > > > > stage? Or does it mean that the average also contains information
> > > > > > > > > > equivalent to variance?
> > > > > > > > >
> > > > > > > > > We can indeed calculate variances and averages of persistence landscapes
> > > > > > > > > because it's a descriptor whose Banach space formalism permits
> > > > > > > > > statistical calculation. The mean landscape itself does not contain
> > > > > > > > > variance information, we would need to specifically calculate
> > > > > > > > > variance. **This is in contrast to the space of graphs where averages and
> > > > > > > > > variances are ill-defined.**
> > > > > > > > >
> > > > > > > > > > Another way of thinking about it might be to provide graph features
> > > > > > > > > > (landscapes) as a framework for users to compare distributions as they
> > > > > > > > > > wish, without limiting the method of comparison between distributions.
> > > > > > > > >
> > > > > > > > > Indeed you can use persistence landscapes as a graph feature to be used
> > > > > > > > > for graph classification, graph distances or other tasks. We show that
> > > > > > > > > they are beneficial for comparing distributions of graphs and use them
> > > > > > > > > as such. This is extremely useful for the important task of evaluating
> > > > > > > > > graph generative models. If we understand your comment correctly, we
> > > > > > > > > could indeed *also* use persistence landscapes to compare other
> > > > > > > > > distributions of objects, but we believe this to be out of scope for
> > > > > > > > > this work.
> > > > > > > > >
> > > > > > > > > Thank you once again for your comments, please let us know if you have
> > > > > > > > > additional questions.
> > > > > > > > >
> > > > > > > > > [1] O'Bray et al., *Evaluation Metrics for Graph Generative Models:
> > > > > > > > > Problems, Pitfalls, and Practical Solutions*, ICLR 2022

---

> > > > > > > > > > ### Comment · Reviewer_juxy · 2023-08-16
> > > > > > > > > > **Re:**
> > > > > > > > > >
> > > > > > > > > > Thank you for your comments.
> > > > > > > > > >
> > > > > > > > > > As for the definition of the distance of the distribution, Wasserstein distance and KL divergence are the most basic methods commonly known when a distance matrix is required. Taking the distance between the means of a distribution as the distance of the distribution is a more classical method, but is not often used because it is known to be inadequate as a comparison of distributions. In this case, however, since you emphasized the distance between means, I mentioned the comparison with that. Even if the data and subject matter are different, a simple analogy should be considered comparative. Due to the weak validation for the objective of calculating the distance between distributions, we proposed a comparison with the simplest method if there is no conventional method to compare. I do not believe that the attitude that if there is no literature in a similar setting, there is no comparison to be made is not an appropriate attitude. If you do not accept this point, then this is a difference of opinion and the decision should be left to the AC. If you are saying that distance calculations such as graph edit distance do not constitute a distance space and therefore cannot compute a meaningful center or balance point, then just answer that way.
> > > > > > > > > >
> > > > > > > > > > Despite the discussion of the calculability of distances between distributions, another argument was presented: interpolation between data. Since you emphasized and mentioned calculating the distance between distributions as the distance between the means of the landscapes, we were discussing the possible ways of calculating the distance between the distributions. But is it not essential to find the distance between distributions? If it is essential to create a method to do so, because in order to make comparisons regarding distributions, the graph data must be embedded in a distance space (with the best possible properties), then please insist on that. I think the argument is also whether the proposed method constructs a distance space that preserves the properties of the data or one that does not preserve the properties but can be formally computed as a distance space. The latter is acceptable, since it is not obvious that there is a graph that has the interpolation of a landscape as a landscape.
> > > > > > > > > >
> > > > > > > > > > Also, I have been discussing the distance between distributions because you had emphatically commented on the distance between distributions. However, the subject of this paper is the evaluation of graph generative models. From this perspective, I believe this paper compared the distribution of the input and reconstruction data sets. Whether a comparison of the input data distribution and the reconstructed data distribution is appropriate as an evaluation of the generative model is debatable, but it may be one way to make a decision. However, if the subject is to be discussed from this perspective, then it should be about reconstruction performance, not a comparison of distributions. There is a large gap between distribution similarity and reconstruction performance. Even if the reconstruction is not properly done, the possibility of similarity in distribution cannot be ruled out. It may be that the permutation test guarantees this point, but since they are not specified, it is impossible to determine.
> > > > > > > > > >
> > > > > > > > > > As a reviewer, if the proposed method includes multiple elements, I would like to clarify the novelty of each element, whether it is essential (it can be substituted by known methods?), the degree of impact on effectiveness, and whether they are adequately guaranteed. However, the arguments to this end does not seem to be overlaping well.

---

> > > > > > > > > > > ### Author Response · Authors · 2023-08-16
> > > > > > > > > > > **Re:**
> > > > > > > > > > >
> > > > > > > > > > > Thank you for your comment. Before addressing  all the issues you raised, we would like to ask for a clarification: according to our understanding, the Wasserstein distance and KL divergence are not applicable in our case since we cannot 'evaluate' them between distributions of graphs. Could you please elaborate?
> > > > > > > > > > >
> > > > > > > > > > > > If you are saying that distance calculations such as graph edit distance do not constitute a distance space and therefore cannot compute a meaningful center or balance point
> > > > > > > > > > > >
> > > > > > > > > > >
> > > > > > > > > > > The graph edit distance leads to a metric space of graphs. Indeed, we do not see how to calculate a (bary)centre there. For instance, the centre point of two graphs that are edit distance X away is *not* a single well-defined graph. The centre point of the persistent landscapes of these two graphs is a single well-defined landscape.
> > > > > > > > > > >
> > > > > > > > > > > > Despite the discussion of the calculability of distances between distributions, another argument was presented: interpolation between data
> > > > > > > > > > > >
> > > > > > > > > > >
> > > > > > > > > > > We believe this to be a misunderstanding. We were pointing out why we believe the graph edit distance is not applicable in that setting: it provides us only with distances, but not with a way to *find* or *construct* the mean graph, whereas if we are only dealing with high-dimensional vectors, we can easily 'interpolate' between them and find their mean.
> > > > > > > > > > >
> > > > > > > > > > > > The latter is acceptable, since it is not obvious that there is a graph that has the interpolation of a landscape as a landscape.
> > > > > > > > > > > >
> > > > > > > > > > >
> > > > > > > > > > > We do not understand this point. Could you please clarify?  Interpolating between two landscapes is indeed a single well-defined landscape.
> > > > > > > > > > >
> > > > > > > > > > > > Due to the weak validation for the objective of calculating the distance between distributions, we proposed a comparison with the simplest method if there is no conventional method to compare.
> > > > > > > > > > > >
> > > > > > > > > > >
> > > > > > > > > > > There *is* a conventional method for comparing distributions of graphs. This is done using descriptor functions with MMD. We evaluate against this approach and show that our method is more expressive, robust and scalable.
> > > > > > > > > > >
> > > > > > > > > > > > Whether a comparison of the input data distribution and the reconstructed data distribution is appropriate as an evaluation of the generative model is debatable, but it may be one way to make a decision.
> > > > > > > > > > > >
> > > > > > > > > > >
> > > > > > > > > > > Indeed it is extremely important to measure the similarity between the input data distribution and the reconstructed data distribution as the graph generative model is constructed to sample from the input data distribution. Having a metric which can measure the extent to which the model achieves this goal is extremely useful. We believe this to be the appropriate course of action, backed up by the current state of the art in graph generative modelling where it is the standard metric for graph generation evaluation [1][2] and is done using descriptor functions combined with MMD [3] which we show improvement over.
> > > > > > > > > > >
> > > > > > > > > > > [1] Vignac et al., *DiGress: Discrete Denoising diffusion for graph generation,* ICLR 2023
> > > > > > > > > > >
> > > > > > > > > > > [2] Martinkus et al., *SPECTRE: Spectral Conditioning Helps to Overcome the Expressivity Limits of One-shot Graph Generators,* ICML 2022
> > > > > > > > > > >
> > > > > > > > > > > [3] O'Bray et al., *Evaluation Metrics for Graph Generative Models: Problems, Pitfalls, and Practical Solutions*, ICLR 2022

---

> > > > > > > > > > > > ### Comment · Reviewer_juxy · 2023-08-17
> > > > > > > > > > > >
> > > > > > > > > > > > Perhaps I misunderstood your first comment, I thought you were stating that you accepted MMD as the distance between the two graphs and that you would show the comparison of the distributions in a different way, Since the comparison of the proposed method and MMD in the paper's description caused the misunderstanding that it was a comparison of the distance between the two graphs, even though it is essentially a comparison as a distribution, it seems your intention is to correct it to be clear that it is a comparison of distributions. If so, there would be no need to compare it to more classical methods such as MMD. If that is the case, then I apologize for the meaningless discussion.
> > > > > > > > > > > > It seems counter-intuitive to me personally to do a comparison of the means of the distributions as an evaluation of the distributions, but is it important to do a permutation test in order to consider it an appropriate comparison of the distributions?

---

> > > > > > > > > > > > > ### Author Response · Authors · 2023-08-17
> > > > > > > > > > > > > **Re:**
> > > > > > > > > > > > >
> > > > > > > > > > > > >
> > > > > > > > > > > > > >caused the misunderstanding that it was a comparison of the distance between the two graphs, even though it is essentially a comparison as a distribution, it seems your intention is to correct it to be clear that it is a comparison of distributions.
> > > > > > > > > > > > >
> > > > > > > > > > > > >  Apologies for the misunderstanding! Indeed, it is our intention be be more clear that it is a comparison of distributions. **We shall be sure to highlight this in a revision**.
> > > > > > > > > > > > >
> > > > > > > > > > > > >  >It seems counter-intuitive to me personally to do a comparison of the means of the distributions as an evaluation of the distributions
> > > > > > > > > > > > >
> > > > > > > > > > > > >  We see your point. We, as well as previous work for graph distribution comparison, compare means of the distributions. Future work could also consider looking at higher-order moments of the distribution which our framework could naturally account for.
> > > > > > > > > > > > >
> > > > > > > > > > > > >  >is it important to do a permutation test in order to consider it an appropriate comparison
> > > > > > > > > > > > >
> > > > > > > > > > > > >  The permutation test is not important for this; it was only used to measure significance. We envisage future work that creates new graph generative models will use our method (without the permutation test) to get a score between their training distribution of graphs and their sampled graphs. This is currently done using descriptor functions combined with MMD. However, we have shown the advantages of our approach over this in terms of expressivity, robustness and scalability.

---

> > > > > > > > > > > > > > ### Comment · Reviewer_juxy · 2023-08-18
> > > > > > > > > > > > > >
> > > > > > > > > > > > > > Thank you for your comment. I was convinced regarding your opinion.
> > > > > > > > > > > > > > I didn't mention this in my initial review, but could you comment on the following points.
> > > > > > > > > > > > > > - Ricci curvature is typically defined using random walk measures, but is there a connection to DeepWalk[1]? This one also yields a vector representation of a random-walk based graph.
> > > > > > > > > > > > > > - Ricci curvature can be thought of as the strength of flow in the direction of easy gathering, but Heat Kernel, which expresses flow in the direction of diffusion, also has similarities, although its characteristics are different. PersLay[2] is designed to train graphs, not to compare distributions, but it create features of graphs using Heat Kernel-based graph filtration. It is possible to simply replace it with the filtration of your method, but will it make a difference?
> > > > > > > > > > > > > >
> > > > > > > > > > > > > > [1]D. Perozzi et al, DeepWalk: online learning of social representations, KDD'14
> > > > > > > > > > > > > >
> > > > > > > > > > > > > > [2]M Carriere et al, PersLay:ANeuralNetworkLayerforPersistenceDiagramsand NewGraphTopologicalSignatures, AISTATS 2020

---

> > > > > > > > > > > > > > > ### Author Response · Authors · 2023-08-18
> > > > > > > > > > > > > > > **Re:**
> > > > > > > > > > > > > > >
> > > > > > > > > > > > > > > >Ricci curvature is typically defined using random walk measures, but is there a connection to DeepWalk?
> > > > > > > > > > > > > > >
> > > > > > > > > > > > > > > DeepWalk uses random walks and optimises a skip-gram model to get a vectorial representation over **nodes**. This is also similar to node2vec [1] and these node embedding methods have been found to be unstable with respect to parameter choices [2]. This is unlike curvature where we get nice *stability* properties.
> > > > > > > > > > > > > > > Additionally, curvature gives a *single value* over **edges** based on an *optimal transport distance* between random walk measures over nodes. This optimal transport distance proves crucial in terms of expressivity; the initial probability measure over indistinguishable nodes cannot distinguish the strongly regular graphs in our dataset. However, Ricci Curvature is able to distinguish some strongly regular graphs which 3-WL cannot. Additionally, by providing a single value rather than a vectorial representation over nodes/edges, we are able to use the curvature as a filtration function which we show is at least as expressive as the curvature itself (Theorem 5) and performs better on our benchmarks.
> > > > > > > > > > > > > > > Additionally, note that our method is fully permutation invariant, unlike node embeddings obtained via implicit matrix factorization such as deepwalk and node2vec [3].
> > > > > > > > > > > > > > >
> > > > > > > > > > > > > > > > Ricci curvature can be thought of as the strength of flow in the direction of easy gathering, but Heat Kernel, which expresses flow in the direction of diffusion, also has similarities, although its characteristics are different. PersLay is designed to train graphs, not to compare distributions, but it create features of graphs using Heat Kernel-based graph filtration. It is possible to simply replace it with the filtration of your method, but will it make a difference?
> > > > > > > > > > > > > > >
> > > > > > > > > > > > > > > Thanks for the interesting suggestion here. Both approaches highlight the potential and utility of persistent homology to capture and communicate important graph features. Although the Perslay experiments focus on graph classification based on a Heat Kernel Signature filtration, their framework supports training generally for any persistent diagram (not only those arising from graphs). Thus one could certainly build persistence diagrams from our curvature filtrations into the Perslay framework. Similarly, we could use Heat Kernel Signature filtrations to compare distributions of graphs using persistence landscapes. However, we propose using curvature measures, as mentioned previously, because of our theoretical and experimental understanding of their stability and expressivity. This would certainly be interesting to investigate in the future, comparing the performance of different types of filtrations in both our and Perslay's framework.
> > > > > > > > > > > > > > >
> > > > > > > > > > > > > > > We thank the reviewer for the related work and references. We shall add these to the manuscript.
> > > > > > > > > > > > > > >
> > > > > > > > > > > > > > > [1] Grover et al., *node2vec: Scalable Feature Learning for Networks,* SIGKDD International Conference on Knowledge Discovery and Data Mining 2016
> > > > > > > > > > > > > > >
> > > > > > > > > > > > > > > [2] Hacker et al., *On the Surprising Behaviour of node2vec,* Proceedings of Topological, Algebraic, and Geometric Learning Workshops 2022
> > > > > > > > > > > > > > >
> > > > > > > > > > > > > > > [3] Srinivasan et al., *On the Equivalence between Positional Node Embeddings and Structural Graph Representations,* ICLR 2020

---

> > > > > > > > > > > > > > > > ### Comment · Reviewer_juxy · 2023-08-19
> > > > > > > > > > > > > > > >
> > > > > > > > > > > > > > > > Thank you for your comments.
> > > > > > > > > > > > > > > > Although the investigation of heat kernel filtration and other filtration systems is an issue for the future, curvature filtration should be stable and usable at the performance levels currently shown. In the final decision, although there are many issues that require further discussion, many of them are due to the difficulties in the field of evaluation of generative models, and we believe that at least one indicator can be used, and that it is highly valuable because it will attract further discussion. I would raise the overall score to be the final answer.

---

> > > > > > > > > > > > > > > > > ### Author Response · Authors · 2023-08-19
> > > > > > > > > > > > > > > > >
> > > > > > > > > > > > > > > > > Thanks for your willingness to engage in this conversation! We are grateful for the clarifications and we believe that this will also help us in the revision of the paper.

---

### Decision · Program_Chairs · 2023-09-21

**Decision:**

Accept (poster)

**Comment:**

The paper addresses a significant, albeit frequently neglected, issue regarding the quantification of dissimilarity between graph distributions. It introduces an innovative approach to assess graph generative models using discrete curvature and topological data analysis. The method's foundation in theory is robust, and I believe it has the potential to stimulate interest in other domains relevant to the conference's audience. Despite an initial dispute between a reviewer and the authors, a comprehensive conversation ultimately clarified the matter. The disagreement appeared to stem primarily from misconceptions related to certain associated concepts, and it was successfully resolved.